# Large Stepsizes Accelerate Gradient Descent for Regularized Logistic Regression

**Jingfeng Wu**[*]
UC Berkeley
uuujf@berkeley.edu

**Pierre Marion**[*†]
Inria, DI ENS, PSL University
pierre.marion@inria.fr

**Peter L. Bartlett**
UC Berkeley & Google DeepMind
peter@berkeley.edu

## Abstract

We study *gradient descent* (GD) with a constant stepsize for $\ell_2$-regularized logistic regression with linearly separable data. Classical theory suggests small stepsizes to ensure monotonic reduction of the optimization objective, achieving exponential convergence in $\widetilde{\mathcal{O}}(\kappa)$ steps with $\kappa$ being the condition number. Surprisingly, we show that this can be *accelerated* to $\widetilde{\mathcal{O}}(\sqrt{\kappa})$ by simply using a large stepsize—for which the objective evolves *nonmonotonically*. The acceleration brought by large stepsizes extends to minimizing the population risk for separable distributions, improving on the best-known upper bounds on the number of steps to reach a near-optimum. Finally, we characterize the largest stepsize for the local convergence of GD, which also determines the global convergence in special scenarios. Our results extend the analysis of Wu et al. (2024) from convex settings with minimizers at infinity to strongly convex cases with finite minimizers.

## 1 Introduction

Machine learning often involves minimizing regularized empirical risk (see, e.g., Shalev-Shwartz and Ben-David, 2014). An iconic case is *logistic regression with $\ell_2$-regularization*, given by

$$\widetilde{\mathcal{L}}(\mathbf{w}) := \mathcal{L}(\mathbf{w}) + \frac{\lambda}{2}\|\mathbf{w}\|^2, \quad \text{where } \mathcal{L}(\mathbf{w}) := \frac{1}{n}\sum_{i=1}^{n} \ln\left(1 + \exp(-y_i \mathbf{x}_i^\top \mathbf{w})\right). \tag{1}$$

Here, $\lambda > 0$ is the regularization hyperparameter, $\mathbf{w} \in \mathbb{H}$ is the trainable parameter, and $(\mathbf{x}_i, y_i) \in \mathbb{H} \times \{\pm 1\}$ for $i = 1, \ldots, n$ are the training data, where $\mathbb{H}$ is a Hilbert space. We consider a generic optimization algorithm, *gradient descent* (GD), defined as

$$\mathbf{w}_{t+1} := \mathbf{w}_t - \eta \nabla \widetilde{\mathcal{L}}(\mathbf{w}_t), \quad t \geq 0, \quad \mathbf{w}_0 \in \mathbb{H}, \tag{GD}$$

where $\eta > 0$ is a constant stepsize and $\mathbf{w}_0$ is an initialization, e.g., $\mathbf{w}_0 = 0$.

This problem is smooth and strongly convex. Classical optimization theory suggests a small stepsize, for which GD decreases the objective $\widetilde{\mathcal{L}}(\mathbf{w}_t)$ *monotonically* (Nesterov, 2018, Section 1.2.3), which we refer to as the *stable* regime. In this regime, GD achieves an $\varepsilon$ error in $\mathcal{O}(\kappa \ln(1/\varepsilon))$ steps, where $\kappa > 1$ is the condition number of the Hessian of $\widetilde{\mathcal{L}}$ (the smoothness parameter divided by the strong convexity parameter). This step complexity is known to be suboptimal and can be improved to $\mathcal{O}(\sqrt{\kappa}\ln(1/\varepsilon))$ when GD is modified by Nesterov's momentum (Nesterov, 2018, Section 2.2).

A recent line of work shows that GD converges even with large stepsizes that lead to *oscillation* (Wu et al., 2024, other related works will be discussed later in Section 1.1). This is known as the *edge of*

---

[*]Equal contribution.
[†]Work done while P.M. was a postdoc at EPFL, visiting the Simons Institute at UC Berkeley.

39th Conference on Neural Information Processing Systems (NeurIPS 2025).

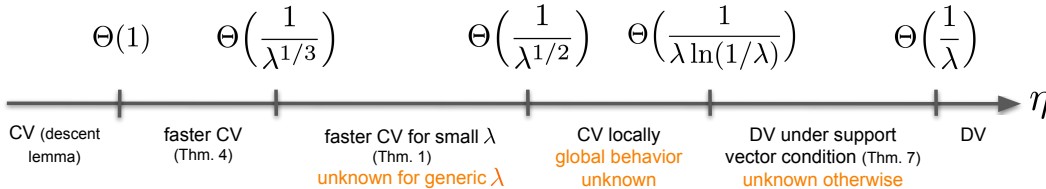

Figure 1: The effect of the stepsize ($\eta$) for GD in logistic regression with $\ell_2$-regularization ($\lambda$). Here, "CV" stands for convergence and "DV" stands for divergence.

*stability* (EoS) (Cohen et al., 2020) regime. Specifically, Wu et al. (2024) considered (unregularized) logistic regression ((1) with $\lambda = 0$) with linearly separable data. Their problem is smooth and convex, but *not* strongly convex. They showed that GD achieves an $\widetilde{\mathcal{O}}(1/\sqrt{\varepsilon})$ step complexity when operating in the EoS regime, which improves the classical $\widetilde{\mathcal{O}}(1/\varepsilon)$ step complexity when operating in the stable regime. However, it is unclear whether large stepsizes would benefit GD in strongly convex problems such as $\ell_2$-regularized logistic regression, for two reasons. First, with linearly separable data, the minimizer of logistic regression is at infinity. Exploiting this property, Wu et al. (2024) showed that GD converges with an arbitrarily large stepsize. However, this is impossible for regularized logistic regression, which is strongly convex and admits a unique, finite minimizer. In this case, GD is *unstable* around the minimizer when the stepsize exceeds a certain threshold (e.g., Hirsch et al., 2013, Section 8), which prevents convergence. Second, Wu et al. (2024) only obtained the accelerated $\widetilde{\mathcal{O}}(1/\sqrt{\varepsilon})$ step complexity for $\varepsilon < 1/n$, where $n$ is the sample size (see their Corollary 2). However, the statistical error (or generalization error) is often larger than $1/n$. In these situations, targeting an optimization error of $\varepsilon < 1/n$ seems less practical, as the statistical error already caps the final population error. It remains unclear whether large stepsizes save computation to minimize population error in the presence of statistical uncertainty.

**Contributions.** We show that large stepsizes accelerate GD for $\ell_2$-regularized logistic regression with linearly separable data, with the following contributions (summarized in Figure 1).

1. For a small regularization hyperparameter ($\lambda = \mathcal{O}(1/n^2)$), we show that GD can achieve an $\varepsilon$ error within $\mathcal{O}(\ln(1/\varepsilon)/\sqrt{\lambda})$ steps. This uses an appropriately large stepsize for which GD operates in the EoS regime. Since the condition number of this problem is $\kappa = \Theta(1/\lambda)$, GD matches the accelerated step complexity of Nesterov's momentum by simply using large stepsizes. We further provide a hard dataset showing that this does not always happen if GD operates in the stable regime.

2. For a general $\lambda$ (independent of $n$), GD still benefits from large stepsizes, achieving an improved step complexity of $\mathcal{O}(\ln(1/\varepsilon)/\lambda^{2/3})$. Assuming a separable data distribution, GD minimizes the (best-known upper bound on) population risk to the statistical bottleneck in $\widetilde{\mathcal{O}}(n^{2/3})$ steps using large stepsizes and regularization. Without one of these, GD takes $\widetilde{\mathcal{O}}(n)$ steps to achieve the same. This improvement provides evidence that large stepsizes accelerate GD under statistical uncertainty.

3. Finally, under additional data assumptions, we derive a critical threshold $\Theta(1/(\lambda \ln(1/\lambda)))$ on the convergent stepsizes for GD in the following sense. With stepsizes that are smaller by a constant factor, GD converges locally (and globally in 1-dimensional cases); with stepsizes that are larger by a constant factor, GD diverges with almost every initialization $\mathbf{w}_0$.

**Terminology.** Formally, we say that GD is in the *stable phase* at step $t$ when $\widetilde{\mathcal{L}}(\mathbf{w}_t)$ decreases monotonically from $t$ onwards, and in the *EoS phase* when it does not. Moreover, we say that a GD run is in the *stable regime* if GD is in the stable phase at the initial step, and in the *EoS regime* if it is in the EoS phase in the beginning but transitions to the stable phase afterward. To give intuition, note that, for a strongly convex and sufficiently differentiable objective, if GD converges, it must enter the stable phase in finite time for a generic initialization. This means that a typical convergent GD run is either in the stable regime or the EoS regime.

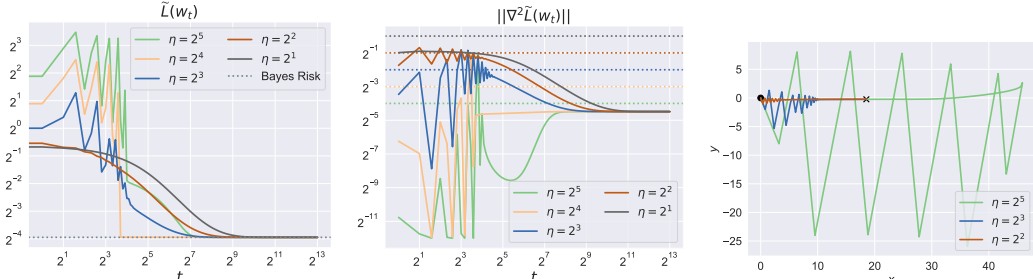

Figure 2: Illustration of large stepsizes accelerating GD. We run constant stepsize GD for an $\ell_2$-regularized logistic regression on a two-dimensional separable dataset. The dataset is given by $\mathbf{x}_1 = (\gamma, 1)$, $\mathbf{x}_2 = (\gamma, -2)$, $y_1 = y_2 = 1$, where $\gamma = 0.2$. The regularization is $\lambda = 2^{-12}$. GD is initialized at $\mathbf{w}_0 = 0$. **Left:** Objective value as a function of training steps. **Middle:** Sharpness (the largest eigenvalue of the Hessian of the objective) as a function of training steps. **Right:** GD trajectory in the parameter space, where the black dot is the GD initialization and the black cross is the minimizer. Additional details and plots are given in Appendix E.

**Simulations.** Our results are illustrated in Figure 2 by running GD for $\ell_2$-regularized logistic regression on a toy two-dimensional separable dataset. Figure 2 suggests that GD converges faster with a larger stepsize by entering the EoS regime, in which the sharpness oscillates around $2/\eta$ in the initial phase.

**Notation.** For two positive-valued functions $f$ and $g$, we write $f \lesssim g$ or $f \gtrsim g$ if there exists $c > 0$ such that for every $x$, $f(x) \leq cg(x)$ or $f(x) \geq cg(x)$, respectively. We write $f \asymp g$ if $f \lesssim g \lesssim f$. We use the standard big-O notation, with $\widetilde{\mathcal{O}}$ and $\widetilde{\Omega}$ to hide polylogarithmic factors within the $\mathcal{O}$ and $\Omega$ notation, respectively. For two vectors $\mathbf{u}$ and $\mathbf{v}$ in a Hilbert space, we denote their inner product by $\langle \mathbf{u}, \mathbf{v} \rangle$ or, equivalently, $\mathbf{u}^\top \mathbf{v}$. We write $\|\mathbf{u}\| := \sqrt{\mathbf{u}^\top \mathbf{u}}$.

## 1.1 Related work

**Edge of stability.** In practice, GD often induces an oscillation yet still converges in the long run (see Wu et al., 2018; Cohen et al., 2020, and references therein). This is referred to by Cohen et al. (2020) as the *edge of stability* (EoS). Since gradient flow would never increase the objective, EoS is essentially a consequence of large stepsizes. Cohen et al. (2020) further pointed out that GD apparently needs to operate in the EoS regime to obtain reasonable optimization and generalization performance in practical deep learning settings. Besides the empirical results, the theoretical mechanism of EoS has been investigated in several papers (see, e.g., Damian et al., 2022; Zhu et al., 2022; Arora et al., 2022, and references therein). In particular, Cohen et al. (2025) proposed a modified ODE called *central flow* to approximate the time-averaged GD trajectory in the EoS regime. Instead of focusing on explaining EoS itself, we study the optimization benefits of GD operating in the EoS regime.

Another line of research has focused on the statistical benefits of large stepsizes for neural networks (see, e.g., Mulayoff et al., 2021; Qiao et al., 2024; Wu et al., 2025b), exploiting the observation that GD with a larger stepsize is constrained to converge to flatter minima (Wu et al., 2018). However, those works assumed the convergence of GD with large stepsizes, which itself is a challenging question. In this regard, our work makes partial progress by showing the global convergence of GD with large stepsizes for $\ell_2$-regularized logistic regression.

**Aggresive stepsize schedulers.** A recent line of research discovered that a variant of GD with certain aggressive stepsize schedulers yields improved convergence for smooth and (strongly) convex optimization (see Altschuler and Parrilo, 2025; Grimmer, 2024; Zhang et al., 2024, and references therein). As a representative example, Altschuler and Parrilo (2025) showed that their GD variant with the *silver stepsize scheduler* attains an improved $\widetilde{\mathcal{O}}(\kappa^{0.7864})$ step complexity for smooth and strongly convex problems with condition number $\kappa$. Similar to our work, they obtained acceleration by using large stepsizes that operate outside the classical stable regime. However, there are several notable differences. First, our problem class, $\ell_2$-regularized logistic regression, is smaller than theirs.

However, we obtain a better $\widetilde{\mathcal{O}}(\kappa^{0.5})$ step complexity. Moreover, we do so with the simpler approach of constant stepsize GD. Finally, from a technical perspective, our analysis is *anytime* as our stepsize choice does not rely on the target error $\varepsilon$ (see Theorem 1), while the algorithm of Altschuler and Parrilo (2025) needs to know the target error $\varepsilon$ in advance.

**Logistic regression.** Logistic regression with linearly separable data is a standard class of problems in optimization and statistical learning theory. For GD with small stepsizes in the stable regime, Soudry et al. (2018) and Ji and Telgarsky (2018) showed that GD diverges to infinity while converging in direction to the maximum $\ell_2$-margin direction. This result was later extended to GD with an arbitrarily large stepsize in the EoS regime (Wu et al., 2023). More recently, Wu et al. (2024) showed that GD with a large stepsize attains an accelerated $\widetilde{\mathcal{O}}(1/\sqrt{\varepsilon})$ step complexity for logistic regression with linearly separable data, demonstrating the benefits of EoS. For the same problem, Zhang et al. (2025) improved the step complexity to $1/\gamma^2$ by considering an *adaptive* large-stepsize variant of GD, where $\gamma$ is the margin of the dataset, and they further showed that this is minimax optimal. As discussed earlier, their results rely strongly on the minimizer being at infinity. In comparison, we focus on logistic regression with $\ell_2$-regularization, where the minimizer is finite.

For $\ell_p$-regularized logistic regression with linearly separable data, Rosset et al. (2004) showed that the regularized empirical risk minimizer converges in direction to the maximum $\ell_p$-margin direction as the regularization tends to zero. Our work complements theirs by considering the step complexity of finding the $\ell_2$-regularized empirical risk minimizer.

For logistic regression with *strictly nonseparable* data, Meng et al. (2024) constructed examples where GD with large stepsizes does not converge globally (even if the stepsize allows local convergence). Similar to our problem, theirs is also smooth, strictly convex, and admits a unique finite minimizer. However, Meng et al. (2024) focused on negative results, while we provide positive results with separable data for the global convergence of GD with large stepsizes. Proving positive results in the nonseparable case is an interesting direction for future work.

## 2 Large stepsizes accelerate GD

We make the following standard assumptions (Novikoff, 1962) throughout the paper.

**Assumption 1** (Bounded and separable data). *Assume the training data $(\mathbf{x}_i, y_i)_{i=1}^n$ satisfies*

A. *for every $i = 1, \ldots, n$, $\|\mathbf{x}_i\| \leq 1$ and $y_i \in \{\pm 1\}$;*

B. *there is a* margin *$\gamma \in (0, 1]$ and a unit vector $\mathbf{w}^*$ such that $y_i \mathbf{x}_i^\top \mathbf{w}^* \geq \gamma$ for every $i = 1, \ldots, n$.*

Under Assumption 1, the objective function $\widetilde{\mathcal{L}}(\cdot)$ defined in (1) is $(1 + \lambda)$-smooth and $\lambda$-strongly convex. The condition number of this problem is $\kappa = \Theta(1/\lambda)$, as the regularization hyperparameter $\lambda$ is typically small. For a small stepsize $\eta = 1/(1 + \lambda) = \Theta(1)$, GD operates in the stable regime, achieving a well-known $\mathcal{O}(\ln(1/\varepsilon)/\lambda)$ step complexity (Nesterov, 2018). Quite surprisingly, we will show that this can be improved to $\mathcal{O}(\ln(1/\varepsilon)/\sqrt{\lambda})$ when the regularization hyperparameter $\lambda$ is small (compared to the reciprocal of the sample size; see Section 2.1), and to $\mathcal{O}(\ln(1/\varepsilon)/\lambda^{2/3})$ for general $\lambda$ (Section 2.2). We obtain this acceleration by using large stepsizes, where GD operates in the EoS regime.

The minimizer, $\mathbf{w}_\lambda := \arg\min \widetilde{\mathcal{L}}(\cdot)$, is unique and finite when $\lambda > 0$.

### 2.1 Matching Nesterov's acceleration under small regularization

Our first theorem characterizes the convergence of GD in the EoS regime when the regularization is small. The proof is deferred to Appendix A.2.

**Theorem 1** (Convergence under small regularization). *Consider (GD) for $\ell_2$-regularized logistic regression (1) under Assumption 1. Assume without loss of generality that $\mathbf{w}_0 = 0$. There exist constants $C_1, C_2, C_3 > 1$ such that the following holds. For every $n \geq 2$,*

$$\lambda \leq \frac{\gamma^2}{C_1 n \ln n} \quad and \quad \eta \leq \min\left\{\frac{\gamma}{\sqrt{C_1 \lambda}}, \frac{\gamma^2}{C_1 n \lambda}\right\},$$

*we have the following:*

- **Phase transition.** *GD must be in the stable phase at step $\tau$ for*

$$\tau := \frac{C_2}{\gamma^2} \max\left\{\eta,\, n,\, \frac{n\ln n}{\eta}\right\},$$

*that is, $\widetilde{\mathcal{L}}(\mathbf{w}_t)$ decreases monotonically for $t \geq \tau$.*

- **The stable phase.** *Moreover, for every $t \geq \tau$, we have*

$$\widetilde{\mathcal{L}}(\mathbf{w}_t) - \min \widetilde{\mathcal{L}} \leq C_3 e^{-\lambda\eta(t-\tau)}, \quad \|\mathbf{w}_t - \mathbf{w}_\lambda\| \leq C_3 \frac{\eta + \ln(\gamma^2/\lambda)}{\gamma} e^{-\lambda\eta(t-\tau)/2}.$$

Theorem 1 provides a convergence guarantee for GD with a stepsize as large as $\eta = \mathcal{O}(1/\sqrt{\lambda})$ (treating other problem-dependent parameters, $\gamma$ and $n$, as constants). With this stepsize, GD might not decrease the objective monotonically—that is, GD might be in the EoS phase at the beginning. Nonetheless, Theorem 1 shows that GD must undergo a phase transition to the stable phase in $\tau = \mathcal{O}(\eta)$ steps. In the stable phase, GD benefits from the large stepsize, achieving an $\varepsilon$ error in $\mathcal{O}\big(\ln(1/\epsilon)/(\eta\lambda)\big)$ subsequent steps.

Theorem 1 recovers the classical $\mathcal{O}(\ln(1/\varepsilon)/\lambda)$ step complexity when GD operates in the stable regime with $\eta = \Theta(1)$. Additionally, Theorem 1 suggests that GD achieves faster convergence in the EoS regime when the stepsize is large, but not larger than $\Theta(1/\sqrt{\lambda})$. Choosing the largest allowed stepsize, GD matches the accelerated step complexity of Nesterov's momentum. This is detailed in the following corollary, with proof deferred to Appendix A.3.

**Corollary 2** (Step complexity under small regularization). *Under the setting of Theorem 1, by using the largest allowed stepsize,*

$$\eta := \min\left\{\frac{\gamma}{\sqrt{C_1\lambda}},\, \frac{\gamma^2}{C_1 n\lambda}\right\},$$

*we have $\widetilde{\mathcal{L}}(\mathbf{w}_t) - \min \widetilde{\mathcal{L}} \leq \varepsilon$ for*

$$t \leq C_4 \max\left\{\frac{1}{\gamma\sqrt{\lambda}},\, \frac{n}{\gamma^2}\right\} \ln(1/\varepsilon),$$

*where $C_4 > 1$ is a constant. Thus for $\lambda \lesssim \gamma^2/n^2$, $\eta \approx 1/\sqrt{\lambda}$ ensures that $t \approx \ln(1/\varepsilon)/\sqrt{\lambda}$ suffices.*

**Matching Nesterov's acceleration.** Treat $\gamma$ as a constant. For a small regularization of $\lambda \lesssim 1/n^2$, Corollary 2 shows that GD achieves a step complexity of $\mathcal{O}(\ln(1/\varepsilon)/\sqrt{\lambda})$ using a large stepsize. Since the condition number is $\kappa = \Theta(1/\lambda)$, this matches the accelerated step complexity of Nesterov's momentum, improving the classical $\mathcal{O}(\ln(1/\varepsilon)/\lambda)$ step complexity for GD in the stable regime.

For a moderately small regularization, $1/n^2 \lesssim \lambda \lesssim 1/(n\ln n)$, Corollary 2 implies a step complexity of $\mathcal{O}(n\ln(1/\varepsilon))$ for GD with a large stepsize, which still improves the classical $\mathcal{O}(\ln(1/\varepsilon)/\lambda)$ step complexity for GD in the stable regime (by at least a logarithmic factor). However, it no longer matches Nesterov's momentum. It is an open question whether large stepsize GD can match Nesterov's momentum for a moderate (or large) regularization.

**A lower bound for stable convergence.** We have shown that large stepsizes accelerate the convergence of GD. This acceleration effect is closely tied to operating in the EoS regime. To clarify, our next theorem shows that GD in the stable regime suffers from an $\widetilde{\Omega}(1/\lambda)$ step complexity in the worst case. Its proof is deferred to Appendix B.

**Theorem 3** (A lower bound). *Consider* (GD) *for $\ell_2$-regularized logistic regression* (1) *with $\mathbf{w}_0 = 0$ and the following dataset (satisfying Assumption 1):*

$$\mathbf{x}_1 = (\gamma,\, 0.9), \quad \mathbf{x}_2 = (\gamma,\, -0.5), \quad y_1 = y_2 = 1, \quad 0 < \gamma < 0.1.$$

*There exist $C_1, C_2, C_3 > 1$ that only depend on $\gamma$ such that the following holds. For every $\lambda < 1/C_1$ and $\varepsilon < C_2\lambda\ln^2(1/\lambda)$, if $\eta$ is such that $(\widetilde{\mathcal{L}}(\mathbf{w}_t))_{t\geq 0}$ is nonincreasing, then*

$$\widetilde{\mathcal{L}}(\mathbf{w}_t) - \min \widetilde{\mathcal{L}} \leq \varepsilon \quad \Rightarrow \quad t \geq \frac{\ln(1/\varepsilon)}{C_3\lambda\ln^2(1/\lambda)}.$$

It is worth noting that Theorem 3 focuses on the common asymptotic case of a small $\varepsilon$; for $\varepsilon \gtrsim \lambda\ln^2(1/\lambda)$, the step complexity is $\Omega(1/\varepsilon)$, which is reflected by its proof in Appendix B.

**A limitation.** We conclude this part by discussing a limitation of our Theorem 1. Note that Theorem 1 only allows a small regularization such that $\lambda \lesssim 1/(n \ln n)$. A regularization of this order might be suboptimal in the presence of statistical noise. Moreover, the proof of Theorem 1 implies that, in the stable phase, all training data are classified correctly (see Lemma 13 in Appendix A.2). Therefore, the allowed regularization is too small to prevent the minimizer from perfectly classifying the training data. A similar limitation is encountered in the prior work of Wu et al. (2024), who showed acceleration with large stepsizes only when targeting an optimization error small enough to imply perfect classification of the training data (see their Corollary 2).

Depending on the statistical model, a perfect fit to the training data does not necessarily lead to overfitting (a phenomenon known as *benign overfitting*, see Bartlett et al., 2020). Even so, a larger regularization such as $\lambda \gtrsim 1/n$ often leads to better performance in many statistical models (one such case will be discussed in Section 3). Below, we show that GD can still benefit from large stepsizes even with regularization larger than $1/(n \ln n)$. This allows large regularization that leads to misclassification of the training data.

## 2.2 Improved convergence under general regularization

Our next theorem characterizes the convergence of GD for $\ell_2$-regularized logistic regression in the EoS regime with a general regularization hyperparameter $\lambda$. The proof is deferred to Appendix A.4.

**Theorem 4** (Convergence under general regularization). *Consider* (GD) *for $\ell_2$-regularized logistic regression* (1) *under Assumption 1. Assume without loss of generality that $\mathbf{w}_0 = 0$. There exist constants $C_1, C_2, C_3 > 1$ such that the following holds. For every*

$$\lambda \leq \frac{\gamma^2}{C_1}, \quad \eta \leq \left(\frac{\gamma^2}{C_1 \lambda}\right)^{1/3},$$

*we have the following:*

- **Phase transition time.** *GD must be in the stable phase at step $\tau := C_2 \max\{1, \eta^2\}/\gamma^2$.*

- **The stable phase.** *Moreover, for $t \geq \tau$, we have*

$$\widetilde{\mathcal{L}}(\mathbf{w}_t) - \min \widetilde{\mathcal{L}} \leq \frac{C_3}{\eta} e^{-\lambda \eta (t-\tau)}, \quad \|\mathbf{w}_t - \mathbf{w}_\lambda\| \leq C_3 \frac{\eta + \ln(\gamma^2/\lambda)}{\gamma} e^{-\lambda \eta (t-\tau)/2}.$$

Similarly to Theorem 1, Theorem 4 allows a large stepsize, in which GD might be in the EoS phase at the beginning, then it must transition to the stable phase in finite steps, achieving an exponential convergence subsequently.

Unlike Theorem 1, where the allowed regularization and phase transition time are functions of the sample size $n$, Theorem 4 is completely independent of the sample size $n$. In particular, it allows for a large regularization of order $1/(n \ln n) \lesssim \lambda \lesssim 1$, with which the minimizer of the regularized logistic regression (1) might not correctly classify the training data.

The relaxation of the allowed regularization is obtained at the price of a tighter constraint on the allowed stepsize, $\eta = \mathcal{O}(1/\lambda^{1/3})$, and a slower phase transition time, $\tau = \Theta(\eta^2)$. Nonetheless, Theorem 4 still implies that large stepsizes lead to acceleration, as explained in the following corollary. Its proof is included in Appendix A.5.

**Corollary 5** (Step complexity under general regularization). *Under the setting of Theorem 4, by using the largest allowed stepsize, $\eta := (\gamma^2/(C_1\lambda))^{1/3}$, we have $\widetilde{\mathcal{L}}(\mathbf{w}_t) - \min \widetilde{\mathcal{L}} \leq \varepsilon$ for*

$$t \leq C_3 \frac{\ln(1/\varepsilon)}{(\gamma \lambda)^{2/3}},$$

*where $C_3 > 1$ is a constant.*

Ignoring the dependence on $\gamma$, Corollary 5 shows that GD achieves an $\varepsilon$ error in $\mathcal{O}(\ln(1/\varepsilon)/\lambda^{2/3})$ steps using a large stepsize. This improves the classical $\mathcal{O}(\ln(1/\varepsilon)/\lambda)$ step complexity for GD in the stable regime, although it does not match Nesterov's momentum.

We remark that the predictions of Corollaries 2 and 5 are incomparable even in the regime where both are applicable, that is, $\lambda \lesssim (n \ln n)^{-1}$. Specifically, in this regime, Corollary 5 predicts a step

complexity of $\widetilde{\mathcal{O}}(\lambda^{-2/3})$ while Corollary 2 predicts a step complexity of $\widetilde{\mathcal{O}}(\max\{\lambda^{-1/2}, n\})$. The prediction of Corollary 5 is worse for $\lambda \lesssim n^{-3/2}$ but is better for $n^{-3/2} \lesssim \lambda \lesssim (n\ln n)^{-1}$. Thus, Corollaries 2 and 5 are incomparable even in the joint applicable regime, suggesting our analysis is improvable. Technically, this mismatch stems from two distinct approaches for analyzing phase transition (see Section 2.3). We leave it as future work to improve our analysis.

In statistical learning contexts, the (optimal) regularization hyperparameter $\lambda$ is often a function of the sample size $n$, for example, $\lambda = \Theta(1/n^\alpha)$ for some $\alpha > 0$. In contrast to Corollary 2, which only applies to small regularization (with $\alpha > 1$), the acceleration implied by Corollary 5 applies to any such $\lambda$ (in particular, with $0 < \alpha \le 1$). Specifically, Corollary 5 shows an $\mathcal{O}(n^{2\alpha/3} \ln(1/\varepsilon))$ step complexity for GD when $\lambda = \Theta(1/n^\alpha)$ for any $\alpha > 0$. We will revisit this later in Section 3 and show the acceleration of large stepsizes in a statistical learning setting.

## 2.3 Technical overview

In this part, we discuss key ideas in our analysis and elaborate on our technical innovations compared to the prior work of Wu et al. (2024).

**Bounds in the EoS phase.** The following lemma provides bounds on the logistic empirical risk and parameter norm for any time; in particular, it applies to the EoS phase.

**Lemma 1** (EoS bounds). *Assume that $\eta\lambda \le 1/2$ and $\mathbf{w}_0 = 0$. Then for every $t$, and in particular in the EoS phase, we have*

$$\frac{1}{t}\sum_{k=0}^{t-1}\mathcal{L}(\mathbf{w}_k) \le 10\frac{\eta^2 + \ln^2(e + \gamma^2\min\{\eta t, 1/\lambda\})}{\gamma^2\min\{\eta t, 1/\lambda\}}, \quad \|\mathbf{w}_t\| \le 4\frac{\eta + \ln(e + \gamma^2\min\{\eta t, 1/\lambda\})}{\gamma}.$$

Lemma 1 recovers the EoS bounds in (Wu et al., 2024) (see their Lemma 8 in Appendix B) for the special case of $\lambda = 0$. The intuition is that, for $\eta t < 1/\lambda$, the regularization term is negligible compared to the logistic term. However, these bounds are too crude for a large $t$.

**New challenges.** The analysis by Wu et al. (2024) relies on the self-boundedness of the logistic loss, $\|\nabla^2\mathcal{L}(\mathbf{w})\| \le \mathcal{L}(\mathbf{w})$, and that the minimizers of $\mathcal{L}(\cdot)$ appear at infinity. Although GD with a large stepsize oscillates initially, it keeps moving towards infinity along the maximum $\ell_2$-margin direction, which reduces the objective $\mathcal{L}(\mathbf{w})$ in the long run. Once GD hits a small objective value, $\mathcal{L}(\mathbf{w}) \lesssim 1/\eta$, it enters the stable phase as the local landscape becomes flat due to the self-boundedness. In the stable phase, GD continues to move towards infinity along the maximum $\ell_2$-margin direction.

However, the situation is significantly different in the presence of an $\ell_2$-regularization. In this case, the minimizer has a small norm, $\|\mathbf{w}_\lambda\| = \mathcal{O}(\ln(1/\lambda))$ (see Lemma 3 in Appendix A.1). But large stepsizes GD can go as far as $\Theta(\eta) = \mathrm{poly}(1/\lambda)$ in the EoS phase (Lemma 1), which is even more distant from the minimizer than the initialization $\mathbf{w}_0 = 0$. Instead of moving towards infinity, in our case, GD must move backwards (if it converges).

Consider a flat region defined as

$$\left\{\mathbf{w} : \|\nabla^2\widetilde{\mathcal{L}}(\mathbf{w})\| = \|\nabla^2\mathcal{L}(\mathbf{w})\| + \lambda \lesssim 1/\eta\right\} \approx \left\{\mathbf{w} : \mathcal{L}(\mathbf{w}) \lesssim 1/\eta\right\}.$$

When GD enters this region, we expect $\widetilde{\mathcal{L}}(\mathbf{w})$ to decrease in the next step (see Lemma 12 in Appendix A.1). However, different from Wu et al. (2024), this does not guarantee that GD stays in this region. In fact, the regularization term leads to contraction towards zero, so a decrease of $\widetilde{\mathcal{L}}(\mathbf{w})$ may cause an increase of $\mathcal{L}(\mathbf{w})$, and then GD might leave this region. In Theorems 1 and 4, we identify two situations where GD stays in the flat region, respectively, as explained below.

**Intuition of Theorem 4.** When $\eta \lesssim 1/\lambda^{1/3}$, Lemma 1 implies a small regularization term throughout the training, $\lambda\|\mathbf{w}_t\|^2 = \widetilde{\mathcal{O}}(\lambda\eta^2) = \widetilde{\mathcal{O}}(1/\eta)$. Thus, the logistic term dominates the whole objective in the EoS phase, and $\mathcal{L}(\mathbf{w}) \lesssim 1/\eta$ is nearly the same as $\widetilde{\mathcal{L}}(\mathbf{w}) \lesssim 1/\eta$. By the decrease of $\widetilde{\mathcal{L}}(\mathbf{w})$ within the flat region, we can show GD stays in this region by induction (see Lemma 15 in Appendix A.4).

Table 1: Step complexities for variants of GD to reach a population risk of $\widetilde{\mathcal{O}}(1/n)$.

| algorithm | # steps | $\lambda$ | $\eta$ | population risk |
|---|---|---|---|---|
| GD | $\mathcal{O}(n)$ | $0$ | $\Theta(1)$ | $\widetilde{\mathcal{O}}(1/(\gamma^2 n))$ |
| | $\mathcal{O}(n \ln n)$ | $1/n$ | $1$ | $\widetilde{\mathcal{O}}(1/(\gamma^2 n))$ |
| | $\mathcal{O}((n/\gamma)^{2/3} \ln n)$ | $1/n$ | $\Theta((\gamma^2 n)^{1/3})$ | $\widetilde{\mathcal{O}}(1/(\gamma^2 n))$ |
| Nesterov's momentum | $\mathcal{O}(n^{1/2} \ln n)$ | $1/n$ | $1$ | $\widetilde{\mathcal{O}}(1/(\gamma^2 n))$ |
| adaptive GD | $\mathcal{O}(1/\gamma^2)$ | $0$ | $\Theta(\ln n)$ | $\widetilde{\mathcal{O}}(1/(\gamma^4 n))$ |

**Intuition of Theorem 1.** For $\eta \lesssim 1/\lambda^{1/2}$, the above arguments no longer work, since the regularization term could be as large as $\widetilde{\mathcal{O}}(\lambda\eta^2) = \widetilde{\mathcal{O}}(1)$ in the EoS phase. Alternatively, we compare the size of the gradients from the logistic term $\|\nabla\mathcal{L}(\mathbf{w})\|$ and the regularization term $\|\lambda\mathbf{w}\|$. If the former is larger, then the logistic term decreases; if the latter is larger, then by the exponential tail of the logistic loss, we conclude that $\mathcal{L}(\mathbf{w}) \approx \|\nabla\mathcal{L}(\mathbf{w})\| \leq \|\lambda\mathbf{w}\| = \widetilde{\mathcal{O}}(1/\eta)$, where the last equality is by Lemma 1. In both cases, GD stays in the flat region (see Lemma 13 in Appendix A.2).

## 3 Benefits of large stepsizes under statistical uncertainty

In this section, we apply Theorem 4 in a statistical learning setting, showing that the acceleration of large stepsizes continues to hold even under statistical uncertainty. We make the following natural assumption on the population data distribution.

**Assumption 2** (Bounded and separable distribution). *Assume that $(\mathbf{x}_i, y_i)_{i=1}^n$ are independent copies of $(\mathbf{x}, y)$ that follows a distribution such that*

A. *the label is binary, $y \in \{\pm 1\}$, and $\|\mathbf{x}\| \leq 1$, almost surely;*

B. *there exist a margin $\gamma > 0$ and a unit vector $\mathbf{w}^*$ such that $y\mathbf{x}^\top \mathbf{w}^* \geq \gamma$, almost surely.*

The population risk of an estimator $\hat{\mathbf{w}}$ is defined as

$$\mathcal{L}_{\text{test}}(\hat{\mathbf{w}}) := \mathbb{E}\ln\left(1 + \exp(-y\mathbf{x}^\top\hat{\mathbf{w}})\right),$$

where the expectation is over the distribution of $(\mathbf{x}, y)$ satisfying Assumption 2.

The following Proposition 6 gives the best-known population risk upper bound (without assuming enormous burn-in samples) in the setting of Assumption 2. This is a direct consequence of the fast rate established by Srebro et al. (2010, Theorem 1) using *local Rademacher complexity* (Bartlett et al., 2005). A variant of Proposition 6 also appears in Schliserman and Koren (2024, Proposition 1). We include its proof in Appendix C.1 for completeness.

**Proposition 6** (A population risk bound). *Suppose that $(\mathbf{x}_i, y_i)_{i=1}^n$ satisfies Assumption 2. Then for every $\hat{\mathbf{w}}$, with probability at least $1 - \delta$ over the randomness of sampling $(\mathbf{x}_i, y_i)_{i=1}^n$, we have*

$$\mathcal{L}_{\text{test}}(\hat{\mathbf{w}}) \leq C\left(\mathcal{L}(\hat{\mathbf{w}}) + \frac{\max\{1, \|\hat{\mathbf{w}}\|^2\}\left(\ln^3(n) + \ln(1/\delta)\right)}{n}\right),$$

*where $C > 1$ is a constant.*

Recall that the minimizer of $\mathcal{L}(\cdot)$ is at infinity under Assumption 2. However, Proposition 6 suggests that a good estimator should balance its fit to the training data (measured by $\mathcal{L}(\hat{\mathbf{w}})$) and its complexity (measured by $\|\hat{\mathbf{w}}\|$). It is also worth noting that the upper bound in Proposition 6 is at least $\widetilde{\Omega}(1/n)$—a bottleneck that stems from the statistical uncertainty. With this in mind, we are ready to discuss the number of steps needed by GD (and its variants) to minimize the population risk to the statistical bottleneck. Table 1 summarizes the results, which we explain in detail below.

**Logistic regression with $\ell_2$-regularization.** Let us first consider the minimizer of the $\ell_2$-regularized logistic regression, $\mathbf{w}_\lambda := \arg\min \widetilde{\mathcal{L}}(\cdot)$. With direct calculation, setting $\lambda = \Theta(1/n)$ minimizes the upper bound in Proposition 6, resulting in an $\widetilde{\mathcal{O}}(1/(\gamma^2 n))$ population risk (see Appendix C.2 for details). This is nearly optimal ignoring the logarithmic factors and dependence on $\gamma$. Clearly, the same bound applies to any approximate minimizer $\hat{\mathbf{w}}$ such that $\|\hat{\mathbf{w}} - \mathbf{w}_\lambda\| \le \varepsilon := 1/\mathrm{poly}(n)$. To obtain such an approximate minimizer,

- GD with a small stepsize $\eta = 1$ needs $\mathcal{O}(n \ln n)$ steps by the classical optimization theory;
- GD with a large stepsize $\eta = \Theta((\gamma^2 n)^{1/3})$ needs $\mathcal{O}((n/\gamma)^{2/3} \ln n)$ steps by Theorem 4;
- Nesterov's momentum needs $\mathcal{O}(n^{1/2} \ln n)$ steps by the classical optimization theory.

These suggest that large stepsizes accelerate GD in the presence of statistical uncertainty, although not as fast as Nesterov's momentum.

**Logistic regression without regularization.** Instead of solving regularized logistic regression, one can also apply GD to the unregularized logistic regression with early stopping to obtain a small population risk. For instance, Shamir (2021) showed that GD with a small stepsize $\eta = 1$ achieves a population risk of $\widetilde{\mathcal{O}}(1/(\gamma^2 n))$ in $\mathcal{O}(n)$ steps. A similar result is obtained by Schliserman and Koren (2024) using a different proof technique.

For GD with a larger stepsize, Wu et al. (2024) obtained an empirical risk bound of $\mathcal{L}(\mathbf{w}_t) = \mathcal{O}\big((\eta^2 + \ln^2(\eta t))/(\gamma^2 \eta t)\big)$ and a parameter norm bound of $\|\mathbf{w}_t\| = \mathcal{O}\big((\eta + \ln(\eta t))/\gamma\big)$. Note that we do not consider their accelerated empirical risk bound here, as it only applies after $\Theta(n)$ steps. Plugging these bounds into Proposition 6, however, one cannot resolve for a stepsize $\eta$ better than the choice of $\eta = 1$ (ignoring logarithmic factors). That is, without regularization, solely using a large stepsize does not accelerate GD in the presence of statistical uncertainty. This sets an interesting gap between our acceleration results and those by Wu et al. (2024).

One can also solve logistic regression via *adaptive* GD (Ji and Telgarsky, 2021; Zhang et al., 2025), defined as $\mathbf{w}_{t+1} := \mathbf{w}_t - \eta \nabla \mathcal{L}(\mathbf{w}_t)/\mathcal{L}(\mathbf{w}_t)$. This is faster for optimization than GD as it adapts to the curvature. Specifically, Zhang et al. (2025) obtained an empirical risk bound of $\mathcal{L}(\bar{\mathbf{w}}_t) \le \exp(-\Theta(\gamma^2 \eta t))$ for $t > 1/\gamma^2$ (see their Theorem 2.1) and a parameter norm bound of $\|\bar{\mathbf{w}}_t\| \le \eta t$ (see the proof of their Theorem 2.2), where $\bar{\mathbf{w}}_t$ is the average of the iterates up to step $t$. Plugging these bounds into Proposition 6, we minimize the upper bound by setting $\eta = \Theta(\ln n)$ and $t = \Theta(1/\gamma^2)$, with which the population risk is $\widetilde{\mathcal{O}}(1/(\gamma^4 n))$. Although the step complexity is much improved, the population risk seems to have a suboptimal dependence on $\gamma$.

**A limitation.** We note that the above discussion is based on the best-known population risk upper bound in a statistical setting specified by Assumption 2. Depending on the actual data distribution, the population risk might be smaller than that (although we suspect the upper bound is nearly sharp in the worst case). We leave it for future work to investigate the effect of large stepsizes in broader statistical learning settings.

## 4 A critical threshold on the convergent stepsizes

We have shown the global convergence of GD with stepsizes as large as $\mathcal{O}(1/\sqrt{\lambda})$ in Section 2. Clearly, if $\eta > 2/\lambda$, GD diverges with almost every initialization. But the largest convergent stepsize is unclear yet—this is the focus of this section. We will show the largest convergent stepsize is $\Theta(1/(\lambda \ln(1/\lambda)))$ under the following technical condition:

**Assumption 3** (Support vectors condition)**.** *Let $\mathcal{S}_+$ be the index set of the support vectors associated with nonzero dual variables (formally defined in Appendix D.1). Assume that $\mathrm{rank}\{\mathbf{x}_i : i \in \mathcal{S}_+\} = \mathrm{rank}\{\mathbf{x}_1, \ldots, \mathbf{x}_n\}$.*

Assumption 3 is widely used in the literature of logistic regression (Soudry et al., 2018; Ji and Telgarsky, 2021; Wu et al., 2023), requiring the support vectors to be generic. Under this condition, the features $(\mathbf{x}_i)$ can be decomposed into a separable component and a strictly nonseparable component (Wu et al., 2023). Under Assumption 3, our next theorem sharply characterizes the largest convergent stepsizes, with proof deferred to Appendix D.1.

**Theorem 7** (The critical stepsize). *Suppose that $\mathbb{H}$ is finite-dimensional and that Assumptions 1 and 3 hold. Consider (GD) for $\ell_2$-regularized logistic regression (1). Let $\eta_{\mathrm{crit}} := 1/(\lambda \ln(1/\lambda))$. Then there exist $C_1, C_2 > 1$ that only depend on the dataset (but not on $\lambda$) such that the following holds. For every $\lambda \leq 1/C_1$, we have*

- *If $\eta \leq \eta_{\mathrm{crit}}/C_2$, then GD converges locally. That is, there exists $r > 0$ such that, for every $\mathbf{w}_0$ satisfying $\|\mathbf{w}_0 - \mathbf{w}_\lambda\| < r$ and every such $\eta$, we have $\widetilde{\mathcal{L}}(\mathbf{w}_t) \to \min \widetilde{\mathcal{L}}$.*

- *If $\eta \geq C_2 \eta_{\mathrm{crit}}$, then GD diverges for almost every $\mathbf{w}_0$. That is, there exists $\varepsilon > 0$ such that, excluding a measure zero set of $\mathbf{w}_0$, we have $\widetilde{\mathcal{L}}(\mathbf{w}_t) - \min \widetilde{\mathcal{L}} > \varepsilon$ for infinitely many $t$.*

Theorem 7 suggests that if the stepsize exceeds the critical threshold $\eta_{\mathrm{crit}}$ by a constant factor, GD must diverge except with a "lucky" initialization. The critical threshold $\eta_{\mathrm{crit}}$ improves the trivial divergent threshold of $2/\lambda$ by a logarithmic factor, and is tight in the sense that GD converges locally for any stepsize smaller than that by a constant factor. This is in sharp contrast to unregularized logistic regression, where GD converges globally for any stepsize (Wu et al., 2023, 2024).

It remains open whether GD converges *globally* with stepsizes of order $1/\sqrt{\lambda} \lesssim \eta \lesssim \eta_{\mathrm{crit}}$. In the special case where $\mathbb{H}$ is 1-dimensional, we provide an affirmative answer in Theorem 8 in Appendix D.2 along with a step complexity of $\mathcal{O}\big(\ln(1/(\varepsilon\lambda\ln(1/\lambda)))/(\eta\lambda)\big)$. We also refer the reader to (Meng et al., 2024, 2025) for a fine-grained convergence analysis in this case. In the general finite-dimensional case, we conjecture that the answer is affirmative in the following sense:

**Conjecture 1.** *Under the setting of Theorem 7, if $\eta \leq \eta_{\mathrm{crit}}/C_2$ and $\mathbf{w}_0$ is sampled uniformly at random from a unit ball, then GD converges with high probability over the randomness of initialization.*

## 5   Concluding remarks

We consider gradient descent (GD) with a constant stepsize applied to $\ell_2$-regularized logistic regression with linearly separable data. We show that, for a small enough regularization, GD can match the acceleration of Nesterov's momentum by simply using an appropriately large stepsize—with which the objective evolves nonmonotonically. Furthermore, we show that this acceleration brought by large stepsizes holds even under statistical uncertainty. Finally, we calculate the largest possible stepsize with which GD can converge (locally).

This work focuses on the cleanest setup with logistic loss and linear predictors. However, the results presented are ready to be extended to other loss functions (Wu et al., 2024), neural networks in the lazy regime (Wu et al., 2024), and two-layer networks with linearly separable data and bi-Lipschitz activation (Cai et al., 2024). We do not foresee significant new technical challenges here.

There are three future directions worth noting. First, in the context of our paper, does GD converge globally with large stepsizes below the proposed critical threshold? Second, is there a natural statistical learning setting such that GD with large stepsizes generalizes better than GD with small ones? Finally, is there a generic optimization theory for the convergence of GD with large stepsizes? Specifically, is there a general framework to prove the convergence of constant stepsize GD without relying on the descent lemma?

## Acknowledgments

We gratefully acknowledge the NSF's support of FODSI through grant DMS-2023505 and of the NSF and the Simons Foundation for the Collaboration on the Theoretical Foundations of Deep Learning through awards DMS-2031883 and #814639 and of the ONR through MURI award N000142112431. P.M. is supported by a Google PhD Fellowship. The authors are grateful to the Simons Institute for the Theory of Computing for hosting them during parts of this work.

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

# A  Upper bounds on convergence in the EoS regime

Throughout this section, we assume Assumption 1 holds; we set $\mathbf{w}_0 = 0$ without loss of generality. Let $\ell(z) := \ln(1 + \exp(-z))$ be the logistic loss. Define the gradient potential as

$$\mathcal{G}(\mathbf{w}) := \frac{1}{n} \sum_{i=1}^{n} |\ell'(y_i \mathbf{x}_i^\top \mathbf{w})| = \frac{1}{n} \sum_{i=1}^{n} \frac{1}{1 + \exp(y_i \mathbf{x}_i^\top \mathbf{w})}.$$

Recall that $\mathbf{w}_\lambda := \arg\min \widetilde{\mathcal{L}}(\cdot)$.

We first establish useful lemmas in Appendix A.1. We then prove our first set of upper bounds, Theorem 1 and Corollary 2, in Appendices A.2 and A.3, respectively. Finally, we prove our second set of upper bounds, Theorem 4 and Corollary 5, in Appendices A.4 and A.5, respectively.

## A.1  Basic lemmas

We begin with the self-boundedness property.

**Lemma 2** (Self-boundedness of the logistic function). *For all $z \in \mathbb{R}$, we have*

$$\ell''(z) < |\ell'(z)| < \ell(z).$$

*Proof of Lemma 2.* First notice that, for $\alpha > 0$,

$$(1 + \alpha) \ln(1 + \alpha) > \alpha. \tag{2}$$

Indeed, the function $J(\alpha) = (1 + \alpha) \ln(1 + \alpha) - \alpha$ satisfies $J(0) = 0$ and $J'(\alpha) = \ln(1 + \alpha)$, which is positive for $\alpha > 0$. Now, since $\ell'(z) = -1/(1 + \exp(z))$, we have

$$\ell''(z) = \frac{\exp(z)}{(1 + \exp(z))^2}$$
$$< \frac{1}{1 + \exp(z)} = |\ell'(z)|$$
$$< \ln(1 + \exp(-z)) = \ell(z),$$

where the last inequality uses (2) with $\alpha = \exp(-z)$. $\qquad\square$

The following lemma provides bounds on the norm and objective value of $\mathbf{w}_\lambda$.

**Lemma 3** (Bounds on the minimizer). *For $\lambda < \gamma^2$, we have*

$$\|\mathbf{w}_\lambda\| \leq \frac{\sqrt{2} + \ln(\gamma^2/\lambda)}{\gamma}, \quad \widetilde{\mathcal{L}}(\mathbf{w}_\lambda) \leq \frac{\lambda(2 + \ln^2(\gamma^2/\lambda))}{2\gamma^2}.$$

*Proof of Lemma 3.* For

$$\mathbf{u} := \frac{\ln(\gamma^2/\lambda)}{\gamma} \mathbf{w}^*,$$

we have, by Assumption 1,

$$\mathcal{L}(\mathbf{u}) \leq \exp(-\gamma\|\mathbf{u}\|) = \frac{\lambda}{\gamma^2}, \quad \|\mathbf{u}\|^2 = \frac{\ln^2(\gamma^2/\lambda)}{\gamma^2}.$$

Then by definition, we have

$$\widetilde{\mathcal{L}}(\mathbf{w}_\lambda) = \mathcal{L}(\mathbf{w}_\lambda) + \frac{\lambda}{2}\|\mathbf{w}_\lambda\|^2 \leq \mathcal{L}(\mathbf{u}) + \frac{\lambda}{2}\|\mathbf{u}\|^2 \leq \frac{\lambda(2 + \ln^2(\gamma^2/\lambda))}{2\gamma^2}.$$

This completes the proof. $\qquad\square$

The following basic facts are due to Assumption 1.

**Lemma 4** (Basic facts). *For all $\mathbf{w}$, we have*

1. $\gamma \mathcal{G}(\mathbf{w}) \leq \langle -\nabla\mathcal{L}(\mathbf{w}), \mathbf{w}^* \rangle \leq \mathcal{G}(\mathbf{w})$.

2. $\gamma \mathcal{G}(\mathbf{w}) \leq \|\nabla\mathcal{L}(\mathbf{w})\| \leq \mathcal{G}(\mathbf{w})$.

3. $\|\nabla^2\mathcal{L}(\mathbf{w})\| \leq \mathcal{G}(\mathbf{w}) \leq \mathcal{L}(\mathbf{w})$.

4. *If $\mathcal{L}(\mathbf{w}) \leq \ln(2)/n$ or $\mathcal{G}(\mathbf{w}) \leq 1/(2n)$, then $\mathcal{L}(\mathbf{w}) \leq 2\mathcal{G}(\mathbf{w})$.*

*Proof of Lemma 4.* Since

$$-\nabla\mathcal{L}(\mathbf{w}) = \frac{1}{n}\sum_{i=1}^{n}\frac{y_i\mathbf{x}_i}{1+\exp(y_i\mathbf{x}_i^\top\mathbf{w})},$$

the first claim is due to $\gamma \leq y_i\mathbf{x}_i^\top\mathbf{w}^* \leq 1$ by Assumption 1. For the second claim, the lower bound is by the first claim, and the upper bound is by the assumption that $\|\mathbf{x}_i\| \leq 1$. The third claim is due to the self-boundedness of the logistic function (Lemma 2) and the assumption that $\|\mathbf{x}_i\| \leq 1$. In the last claim, both conditions imply that all data are correctly classified, then the claim follows from the fact that $\ln(1+e^{-t}) \leq e^{-t} \leq 2/(1+e^t)$ for $t \geq 0$. $\qquad\square$

The following lemma suggests that GD aligns well with $\mathbf{w}^*$ throughout the training.

**Lemma 5** (Parameter angle). *For $\lambda\eta < 1$, we have*

$$\langle \mathbf{w}_t, \mathbf{w}^* \rangle > 0.$$

*Proof of Lemma 5.* Unrolling (GD) from $\mathbf{w}_0 = 0$, we get

$$\mathbf{w}_t = \sum_{k=0}^{t-1}(1-\eta\lambda)^{t-1-k}(-\eta\nabla\mathcal{L}(\mathbf{w}_k)).$$

So we have

$$\langle \mathbf{w}_t, \mathbf{w}^* \rangle = \sum_{k=0}^{t-1}(1-\eta\lambda)^{t-1-k}\langle -\eta\nabla\mathcal{L}(\mathbf{w}_k), \mathbf{w}^* \rangle \geq \gamma\eta\sum_{k=0}^{t-1}(1-\eta\lambda)^{t-1-k}\mathcal{G}(\mathbf{w}_k) > 0,$$

where the first inequality is by Lemma 4. This completes the proof. $\qquad\square$

The following two lemmas are variants of the split optimization lemma introduced by Wu et al. (2024).

**Lemma 6** (Split optimization, version 1). *Let $\mathbf{u} = \mathbf{u}_1 + \mathbf{u}_2 + \mathbf{u}_3$ for*

$$\mathbf{u}_2 = \frac{\eta}{\gamma}\mathbf{w}^*, \quad \mathbf{u}_1 = \|\mathbf{u}_1\|\mathbf{w}^*, \quad \mathbf{u}_3 = \|\mathbf{u}_3\|\mathbf{w}^*.$$

*For $\lambda\eta < 1$, we have*

$$\frac{\|\mathbf{w}_t - \mathbf{u}\|^2}{2\eta t} + \frac{\gamma\|\mathbf{u}_3\|}{t}\sum_{k=0}^{t-1}\mathcal{G}(\mathbf{w}_k) + \frac{1}{t}\sum_{k=0}^{t-1}\mathcal{L}(\mathbf{w}_k) \leq \mathcal{L}(\mathbf{u}_1) + \frac{\|\mathbf{u}\|^2}{2\eta t} + \frac{\lambda}{t}\sum_{k=0}^{t-1}\langle \mathbf{w}_k, \mathbf{u} \rangle.$$

*Proof of Lemma 6.* We use an extended version of the split optimization technique by Wu et al. (2024), which involves three comparators.

$$\begin{aligned}
\|\mathbf{w}_{t+1} - \mathbf{u}\|^2 &= \|\mathbf{w}_t - \mathbf{u}\|^2 + 2\eta\langle \nabla\widetilde{\mathcal{L}}(\mathbf{w}_t), \mathbf{u} - \mathbf{w}_t \rangle + \eta^2\|\nabla\widetilde{\mathcal{L}}(\mathbf{w}_t)\|^2 \\
&= \|\mathbf{w}_t - \mathbf{u}\|^2 + 2\eta\langle \nabla\mathcal{L}(\mathbf{w}_t) + \lambda\mathbf{w}_t, \mathbf{u} - \mathbf{w}_t \rangle + \eta^2\|\nabla\mathcal{L}(\mathbf{w}_t) + \lambda\mathbf{w}_t\|^2 \\
&\leq \|\mathbf{w}_t - \mathbf{u}\|^2 + 2\eta\langle \nabla\mathcal{L}(\mathbf{w}_t) + \lambda\mathbf{w}_t, \mathbf{u} - \mathbf{w}_t \rangle + 2\eta^2\|\nabla\mathcal{L}(\mathbf{w}_t)\|^2 + 2\eta^2\lambda^2\|\mathbf{w}_t\|^2 \\
&\leq \|\mathbf{w}_t - \mathbf{u}\|^2 + 2\eta\langle \nabla\mathcal{L}(\mathbf{w}_t), \mathbf{u} - \mathbf{w}_t \rangle + 2\eta\lambda\langle \mathbf{w}_t, \mathbf{u} \rangle + 2\eta^2\|\nabla\mathcal{L}(\mathbf{w}_t)\|^2,
\end{aligned}$$

where the last inequality is because $\lambda\eta < 1$. The choice of $\mathbf{u}_2$ and Lemma 4, parts 1 and 2 imply

$$2\eta\langle -\nabla\mathcal{L}(\mathbf{w}_t), \mathbf{u}_2 \rangle \geq 2\eta^2\mathcal{G}(\mathbf{w}_t) \geq 2\eta^2\|\nabla\mathcal{L}(\mathbf{w}_t)\| \geq 2\eta^2\|\nabla\mathcal{L}(\mathbf{w}_t)\|^2.$$

(See also the proof of Lemma 7 in (Wu et al., 2024).) Then we have

$$\|\mathbf{w}_{t+1} - \mathbf{u}\|^2 \le \|\mathbf{w}_t - \mathbf{u}\|^2 + 2\eta\langle\nabla\mathcal{L}(\mathbf{w}_t), \mathbf{u}_1 - \mathbf{w}_t\rangle + 2\eta\langle\nabla\mathcal{L}(\mathbf{w}_t), \mathbf{u}_3\rangle + 2\eta\lambda\langle\mathbf{w}_t, \mathbf{u}\rangle.$$

By convexity and Lemma 4 part 1, we have

$$\|\mathbf{w}_{t+1} - \mathbf{u}\|^2 \le \|\mathbf{w}_t - \mathbf{u}\|^2 + 2\eta\big(\mathcal{L}(\mathbf{u}_1) - \mathcal{L}(\mathbf{w}_t)\big) - 2\eta\gamma\|\mathbf{u}_3\|\mathcal{G}(\mathbf{w}_t) + 2\eta\lambda\langle\mathbf{w}_t, \mathbf{u}\rangle.$$

Telescoping the sum, using $\mathbf{w}_0 = 0$, and rearranging, we get

$$\frac{\|\mathbf{w}_t - \mathbf{u}\|^2}{2\eta t} + \frac{\gamma\|\mathbf{u}_3\|}{t}\sum_{k=0}^{t-1}\mathcal{G}(\mathbf{w}_k) + \frac{1}{t}\sum_{k=0}^{t-1}\mathcal{L}(\mathbf{w}_k) \le \mathcal{L}(\mathbf{u}_1) + \frac{\|\mathbf{u}\|^2}{2\eta t} + \lambda\left\langle\frac{1}{t}\sum_{k=0}^{t-1}\mathbf{w}_k, \mathbf{u}\right\rangle,$$

which completes the proof. □

**Lemma 7** (Split optimization, version 2). *Let $\mathbf{u} = \mathbf{u}_1 + \mathbf{u}_2$ for*

$$\mathbf{u}_2 = \frac{\eta}{2\gamma(1-\eta\lambda)}\mathbf{w}^*, \quad \mathbf{u}_1 = \|\mathbf{u}_1\|\mathbf{w}^*.$$

*For $\lambda\eta < 1$, we have*

$$\|\mathbf{w}_t - \mathbf{u}\|^2 \le (1-\eta\lambda)^t\|\mathbf{u}\|^2 + 2\eta\sum_{k=0}^{t-1}(1-\eta\lambda)^{t-1-k}\Big((1-\eta\lambda)\big(\mathcal{L}(\mathbf{u}_1) - \mathcal{L}(\mathbf{w}_k)\big) + \lambda\|\mathbf{u}\|^2\Big).$$

*Proof of Lemma 7.* Recall that

$$\mathbf{w}_{t+1} - \mathbf{u} = (1-\eta\lambda)(\mathbf{w}_t - \mathbf{u}) - \eta\lambda\mathbf{u} - \eta\nabla\mathcal{L}(\mathbf{w}_t).$$

Taking the squared norm and expanding, we have

$$
\begin{aligned}
\|\mathbf{w}_{t+1} - \mathbf{u}\|^2 &= (1-\eta\lambda)^2\|\mathbf{w}_t - \mathbf{u}\|^2 + 2\eta(1-\eta\lambda)\langle\nabla\mathcal{L}(\mathbf{w}_t), \mathbf{u} - \mathbf{w}_t\rangle + \eta^2\|\nabla\mathcal{L}(\mathbf{w}_t)\|^2 \\
&\quad + \eta^2\lambda^2\|\mathbf{u}\|^2 + 2\eta\lambda(1-\eta\lambda)\langle\mathbf{u}, \mathbf{u} - \mathbf{w}_t\rangle + 2\eta^2\lambda\langle\mathbf{u}, \nabla\mathcal{L}(\mathbf{w}_t)\rangle \\
&= (1-\eta\lambda)^2\|\mathbf{w}_t - \mathbf{u}\|^2 + 2\eta(1-\eta\lambda)\langle\nabla\mathcal{L}(\mathbf{w}_t), \mathbf{u} - \mathbf{w}_t\rangle + \eta^2\|\nabla\mathcal{L}(\mathbf{w}_t)\|^2 \\
&\quad + \eta\lambda(2-\eta\lambda)\|\mathbf{u}\|^2 - 2\eta\lambda(1-\eta\lambda)\langle\mathbf{u}, \mathbf{w}_t\rangle + 2\eta^2\lambda\langle\mathbf{u}, \nabla\mathcal{L}(\mathbf{w}_t)\rangle.
\end{aligned}
$$

The sum of the last two terms of the previous identity is negative,

$$-2\eta\lambda(1-\eta\lambda)\langle\mathbf{u}, \mathbf{w}_t\rangle + 2\eta^2\lambda\langle\mathbf{u}, \nabla\mathcal{L}(\mathbf{w}_t)\rangle = -2\eta\lambda\langle\mathbf{u}, \mathbf{w}_{t+1}\rangle < 0,$$

where the last inequality is by Lemma 5. Moreover, the choice of $\mathbf{u}_2$ and Lemma 4, parts 1 and 2 imply that (see also the proof of Lemma 7 in (Wu et al., 2024))

$$2\eta(1-\eta\lambda)\langle\nabla\mathcal{L}(\mathbf{w}_t), \mathbf{u}_2\rangle + \eta^2\|\nabla\mathcal{L}(\mathbf{w}_t)\|^2 \le 0.$$

So we have

$$
\begin{aligned}
\|\mathbf{w}_{t+1} - \mathbf{u}\|^2 &\le (1-\eta\lambda)^2\|\mathbf{w}_t - \mathbf{u}\|^2 + 2\eta(1-\eta\lambda)\langle\nabla\mathcal{L}(\mathbf{w}_t), \mathbf{u}_1 - \mathbf{w}_t\rangle + \eta\lambda(2-\eta\lambda)\|\mathbf{u}\|^2 \\
&\le (1-\eta\lambda)^2\|\mathbf{w}_t - \mathbf{u}\|^2 + 2\eta(1-\eta\lambda)\big(\mathcal{L}(\mathbf{u}_1) - \mathcal{L}(\mathbf{w}_t)\big) + \eta\lambda(2-\eta\lambda)\|\mathbf{u}\|^2 \\
&\le (1-\eta\lambda)\|\mathbf{w}_t - \mathbf{u}\|^2 + 2\eta(1-\eta\lambda)\big(\mathcal{L}(\mathbf{u}_1) - \mathcal{L}(\mathbf{w}_t)\big) + 2\eta\lambda\|\mathbf{u}\|^2.
\end{aligned}
$$

Unrolling the recursion, we get

$$\|\mathbf{w}_t - \mathbf{u}\|^2 \le (1-\eta\lambda)^t\|\mathbf{u}\|^2 + 2\eta\sum_{k=0}^{t-1}(1-\eta\lambda)^{t-1-k}\Big((1-\eta\lambda)\big(\mathcal{L}(\mathbf{u}_1) - \mathcal{L}(\mathbf{w}_k)\big) + \lambda\|\mathbf{u}\|^2\Big).$$

This completes the proof. □

Based on these split optimization bounds, the following three lemmas establish bounds on parameter norm, gradient potential, and the logistic empirical risk, respectively.

**Lemma 8** (A parameter bound). *For $\eta\lambda \le 1/2$, we have*

$$\|\mathbf{w}_t\| \le 4\frac{\eta + \ln(e + \gamma^2\min\{\eta t, 1/\lambda\})}{\gamma}.$$

*Proof of Lemma 8.* By Lemma 7, we have

$$\|\mathbf{w}_t - \mathbf{u}\|^2 \le (1 - \eta\lambda)^t \|\mathbf{u}\|^2 + \left(2\eta \sum_{k=0}^{t-1} (1-\eta\lambda)^k\right) \left((1-\eta\lambda)\mathcal{L}(\mathbf{u}_1) + \lambda\|\mathbf{u}\|^2\right)$$

$$= (1-\eta\lambda)^t \|\mathbf{u}\|^2 + 2\frac{1-(1-\eta\lambda)^t}{\lambda}\left((1-\eta\lambda)\mathcal{L}(\mathbf{u}_1) + \lambda\|\mathbf{u}\|^2\right)$$

$$\le 2\frac{1-(1-\eta\lambda)^t}{\lambda}\mathcal{L}(\mathbf{u}_1) + 2\|\mathbf{u}\|^2$$

$$\le 2\min\{\eta t, 1/\lambda\}\mathcal{L}(\mathbf{u}_1) + 2\|\mathbf{u}\|^2.$$

In the final inequality, the proof that $1 - (1-\eta\lambda)^t \le \eta\lambda t$ is by induction. For $\eta\lambda \le 1/2$ and

$$\mathbf{u}_1 = \frac{\ln(e + \gamma^2 \min\{\eta t, 1/\lambda\})}{\gamma}\mathbf{w}^*, \quad \mathbf{u}_2 = \frac{\eta}{2\gamma(1-\eta\lambda)}\mathbf{w}^*,$$

we have

$$\|\mathbf{u}_2\| \le \frac{\eta}{\gamma}, \quad \mathcal{L}(\mathbf{u}_1) \le \exp(-\gamma\|\mathbf{u}_1\|) \le \frac{1}{\gamma^2 \min\{\eta t, 1/\lambda\}}.$$

Combining, we have

$$\|\mathbf{w}_t\| \le \|\mathbf{w}_t - \mathbf{u}\| + \|\mathbf{u}\|$$

$$\le \sqrt{2\min\{\eta t, 1/\lambda\}\mathcal{L}(\mathbf{u}_1)} + \left(\sqrt{2}+1\right)\|\mathbf{u}\|$$

$$\le \frac{\sqrt{2}}{\gamma} + \left(\sqrt{2}+1\right)\frac{\eta + \ln(e + \gamma^2 \min\{\eta t, 1/\lambda\})}{\gamma}$$

$$\le 4\frac{\eta + \ln(e + \gamma^2 \min\{\eta t, 1/\lambda\})}{\gamma}.$$

This completes the proof. □

**Lemma 9** (A gradient potential bound)**.** *For $\eta\lambda \le 1/2$, we have*

$$\frac{1}{t}\sum_{k=0}^{t-1} \mathcal{G}(\mathbf{w}_k) \le 11\frac{\eta + \ln(e + \gamma^2 \min\{\eta t, 1/\lambda\})}{\gamma^2 \min\{\eta t, 1/\lambda\}}.$$

*Proof of Lemma 9.* Let $\mathbf{u} = \mathbf{u}_1 + \mathbf{u}_2 + \mathbf{u}_3$ and

$$\mathbf{u}_1 = \frac{\ln(e + \gamma^2 \min\{\eta t, 1/\lambda\})}{\gamma}\mathbf{w}^*, \quad \mathbf{u}_2 = \frac{\eta}{\gamma}\mathbf{w}^*, \quad \mathbf{u}_3 = \frac{\eta + \ln(e + \gamma^2 \min\{\eta t, 1/\lambda\})}{\gamma}\mathbf{w}^*.$$

Then we have

$$\|\mathbf{u}_3\| \ge \frac{1}{\gamma}, \quad \|\mathbf{u}\| = 2\|\mathbf{u}_3\|, \quad \mathcal{L}(\mathbf{u}_1) \le \frac{1}{\gamma^2 \min\{\eta t, 1/\lambda\}}.$$

Moreover, Lemma 8 yields

$$\max_{k \le t} \|\mathbf{w}_k\| \le 4\|\mathbf{u}_3\|.$$

Using Lemma 6, we have

$$\frac{1}{t}\sum_{k=0}^{t-1} \mathcal{G}(\mathbf{w}_k) \le \frac{1}{\gamma\|\mathbf{u}_3\|}\left(\mathcal{L}(\mathbf{u}_1) + \frac{\|\mathbf{u}\|^2}{2\eta t} + \lambda\|\mathbf{u}\| \max_{k \le t} \|\mathbf{w}_k\|\right)$$

$$\le \frac{1}{\gamma}\left(\frac{1}{\|\mathbf{u}_3\|\gamma^2 \min\{\eta t, 1/\lambda\}} + \frac{2\|\mathbf{u}_3\|}{\eta t} + 8\lambda\|\mathbf{u}_3\|\right)$$

$$\le \frac{1}{\gamma}\left(\frac{1}{\gamma \min\{\eta t, 1/\lambda\}} + \left(\frac{2}{\eta t} + 8\lambda\right)\frac{\eta + \ln(e + \gamma^2 \min\{\eta t, 1/\lambda\})}{\gamma}\right)$$

$$\le 11\frac{\eta + \ln(e + \gamma^2 \min\{\eta t, 1/\lambda\})}{\gamma^2 \min\{\eta t, 1/\lambda\}}.$$

This completes the proof. □

**Lemma 10** (A logistic empirical risk bound). *For $\eta\lambda \le 1/2$, we have*

$$\frac{1}{t}\sum_{k=0}^{t-1}\mathcal{L}(\mathbf{w}_k) \le 10\frac{\eta^2 + \ln^2(e + \gamma^2\min\{\eta t, 1/\lambda\})}{\gamma^2\min\{\eta t, 1/\lambda\}}.$$

*Proof of Lemma 10.* Let $\mathbf{u} = \mathbf{u}_1 + \mathbf{u}_2 + \mathbf{u}_3$ with

$$\mathbf{u}_1 = \frac{\ln(e + \gamma^2\min\{\eta t, 1/\lambda\})}{\gamma}\mathbf{w}^*, \quad \mathbf{u}_2 = \frac{\eta}{\gamma}\mathbf{w}^*, \quad \mathbf{u}_3 = 0.$$

Then we have

$$\mathcal{L}(\mathbf{u}_1) \le \frac{1}{\gamma^2\min\{\eta t, 1/\lambda\}}, \quad \|\mathbf{u}\| = \frac{\eta + \ln(e + \gamma^2\min\{\eta t, 1/\lambda\})}{\gamma}.$$

By Lemma 8, we have

$$\max_{k\le t}\|\mathbf{w}_k\| \le 4\|\mathbf{u}\|.$$

Using Lemma 6, we have

$$\frac{1}{t}\sum_{k=0}^{t-1}\mathcal{L}(\mathbf{w}_k) \le \mathcal{L}(\mathbf{u}_1) + \frac{\|\mathbf{u}\|^2}{2\eta t} + \lambda\|\mathbf{u}\|\max_{k\le t}\|\mathbf{w}_k\|$$

$$\le \mathcal{L}(\mathbf{u}_1) + \left(\frac{1}{2\eta t} + 4\lambda\right)\|\mathbf{u}\|^2$$

$$\le \frac{1}{\gamma^2\min\{\eta t, 1/\lambda\}} + 2\left(\frac{1}{2\eta t} + 4\lambda\right)\frac{\eta^2 + \ln^2(e + \gamma^2\min\{\eta t, 1/\lambda\})}{\gamma^2}$$

$$\le 10\frac{\eta^2 + \ln^2(e + \gamma^2\min\{\eta t, 1/\lambda\})}{\gamma^2\min\{\eta t, 1/\lambda\}}.$$

This completes the proof. $\qquad\square$

The next lemma shows that when the gradient potential is small, it remains small under one step of GD (even when the stepsize is large).

**Lemma 11** (Small gradient potential). *Assume that*

$$\lambda \le \frac{1}{3\eta\ln(e + \eta)}.$$

*If in the $t$-th step we have*

$$\mathcal{G}(\mathbf{w}_t) \le \frac{1}{2e^2\eta},$$

*then for every $\mathbf{v}$ in the line segment between $\mathbf{w}_t$ and $\mathbf{w}_{t+1}$, we have*

$$\mathcal{G}(\mathbf{v}) \le \frac{1}{2\eta}.$$

*Proof of Lemma 11.* There exists an $\alpha \in [0, 1]$ such that $\mathbf{v} = \alpha\mathbf{w}_{t+1} + (1 - \alpha)\mathbf{w}_t$. Then for every $1 \le i \le n$, we have

$$y_i\mathbf{x}_i^\top\mathbf{v} = y_i\mathbf{x}_i^\top\left(\alpha((1 - \eta\lambda)\mathbf{w}_t - \eta\nabla\mathcal{L}(\mathbf{w}_t)) + (1 - \alpha)\mathbf{w}_t\right)$$

$$= (1 - \alpha\lambda\eta)y_i\mathbf{x}_i^\top\mathbf{w}_t - \alpha\eta y_i\mathbf{x}_i^\top\nabla\mathcal{L}(\mathbf{w}_t)$$

$$\ge (1 - \alpha\lambda\eta)y_i\mathbf{x}_i^\top\mathbf{w}_t - \eta\|\nabla\mathcal{L}(\mathbf{w}_t)\|$$

$$\ge (1 - \alpha\lambda\eta)y_i\mathbf{x}_i^\top\mathbf{w}_t - \eta\mathcal{G}(\mathbf{w}_t)$$

$$\ge (1 - \alpha\lambda\eta)y_i\mathbf{x}_i^\top\mathbf{w}_t - 1,$$

where the second inequality is by Lemma 4 and the last inequality is because $\mathcal{G}(\mathbf{w}_t) \leq 1/\eta$. So we have

$$\mathcal{G}(\mathbf{v}) \leq \frac{1}{n}\sum_{i=1}^{n}\frac{1}{1+\exp((1-\alpha\lambda\eta)y_i\mathbf{x}_i^\top\mathbf{w}_t - 1)}$$

$$\leq \frac{e}{n}\sum_{i=1}^{n}\frac{1}{1+\exp((1-\alpha\lambda\eta)y_i\mathbf{x}_i^\top\mathbf{w}_t)}$$

$$= \frac{e}{n}\sum_{i=1}^{n}\frac{1}{1+\exp(y_i\mathbf{x}_i^\top\mathbf{w}_t)^{1-\alpha\lambda\eta}}\ .$$

Recall the following inequality:

$$1+x^\beta \geq (1+x)^\beta \quad \text{for } x \geq 0 \text{ and } 0 < \beta < 1.$$

We see this by verifying that the function $x \mapsto 1+x^\beta - (1+x)^\beta$ is increasing for $x > 0$ and maps $0$ to $0$. Applying this inequality and the concavity of the function $x \mapsto x^\beta$ for $0 < \beta < 1$ and $x > 0$, we obtain

$$\mathcal{G}(\mathbf{v}) \leq \frac{e}{n}\sum_{i=1}^{n}\left(\frac{1}{1+\exp(y_i\mathbf{x}_i^\top\mathbf{w}_t)}\right)^{1-\alpha\lambda\eta}$$

$$\leq e\left(\frac{1}{n}\sum_{i=1}^{n}\frac{1}{1+\exp(y_i\mathbf{x}_i^\top\mathbf{w}_t)}\right)^{1-\alpha\lambda\eta}$$

$$= e\mathcal{G}(\mathbf{w}_t)^{1-\alpha\lambda\eta}.$$

Using the assumption on $\mathcal{G}(\mathbf{w}_t)$, we get

$$\mathcal{G}(\mathbf{v}) \leq e\left(\frac{1}{2e^2\eta}\right)^{(1-\alpha\lambda\eta)} = \frac{1}{2e\eta}\left(2e^2\eta\right)^{\alpha\lambda\eta} = \frac{1}{2e\eta}\exp\left(\alpha\lambda\eta\ln(2e^2\eta)\right) \leq \frac{1}{2\eta},$$

where the last inequality is because $\alpha\lambda\eta\ln(2e^2\eta) \leq 1$, which is verified by discussing two cases. If $2e^2\eta \leq 1$, this is trivial; If $2e^2\eta > 1$, this follows from our assumption on $\lambda$ and $\alpha \leq 1$:

$$\alpha\lambda\eta\ln(2e^2\eta) \leq \lambda\eta\ln(2e^2\eta) \leq \frac{\ln(2e^2\eta)}{3\ln(e+\eta)} \leq 1.$$

This completes the proof. $\qquad\square$

The following lemma shows that if the gradient potential is small, then the objective value decreases after one step of GD.

**Lemma 12** (One contraction step). *Assume that*

$$\lambda \leq \frac{1}{3\eta\ln(e+\eta)}.$$

*If in the $t$-th step we have*

$$\mathcal{G}(\mathbf{w}_t) \leq \frac{1}{2e^2\eta},$$

*then we have*

$$\widetilde{\mathcal{L}}(\mathbf{w}_{t+1}) \leq \widetilde{\mathcal{L}}(\mathbf{w}_t) - \frac{\eta}{2}\|\nabla\widetilde{\mathcal{L}}(\mathbf{w}_t)\|^2.$$

*Furthermore, we have*

$$\widetilde{\mathcal{L}}(\mathbf{w}_{t+1}) - \min\widetilde{\mathcal{L}} \leq (1-\eta\lambda)\big(\widetilde{\mathcal{L}}(\mathbf{w}_t) - \min\widetilde{\mathcal{L}}\big) \ \text{and} \ \|\mathbf{w}_{t+1} - \mathbf{w}_\lambda\|^2 \leq (1-\eta\lambda)\|\mathbf{w}_t - \mathbf{w}_\lambda\|^2.$$

*Proof of Lemma 12.* There exists $\mathbf{v}$ in the line segment between $\mathbf{w}_t$ and $\mathbf{w}_{t+1}$ such that

$$\widetilde{\mathcal{L}}(\mathbf{w}_{t+1}) = \widetilde{\mathcal{L}}(\mathbf{w}_t) + \langle\nabla\widetilde{\mathcal{L}}(\mathbf{w}_t), \mathbf{w}_{t+1} - \mathbf{w}_t\rangle + \frac{1}{2}\langle\nabla^2\widetilde{\mathcal{L}}(\mathbf{v}), (\mathbf{w}_{t+1} - \mathbf{w}_t)^{\otimes 2}\rangle$$

$$\leq \widetilde{\mathcal{L}}(\mathbf{w}_t) - \eta \|\nabla \widetilde{\mathcal{L}}(\mathbf{w}_t)\|^2 \left(1 - \frac{\eta \|\nabla^2 \widetilde{\mathcal{L}}(\mathbf{v})\|}{2}\right)$$

$$= \widetilde{\mathcal{L}}(\mathbf{w}_t) - \eta \|\nabla \widetilde{\mathcal{L}}(\mathbf{w}_t)\|^2 \left(1 - \frac{\eta(\lambda + \|\nabla^2 \mathcal{L}(\mathbf{v})\|)}{2}\right).$$

Our assumption on $\lambda$ implies that $\lambda\eta \leq 1/2$. Then by Lemmas 4 and 11, we have

$$\frac{\eta(\lambda + \|\nabla^2 \mathcal{L}(\mathbf{v})\|)}{2} \leq \frac{\eta(\lambda + \mathcal{G}(\mathbf{v}))}{2} \leq \frac{0.5 + 0.5}{2} = \frac{1}{2},$$

which leads to

$$\widetilde{\mathcal{L}}(\mathbf{w}_{t+1}) \leq \widetilde{\mathcal{L}}(\mathbf{w}_t) - \eta \|\nabla \widetilde{\mathcal{L}}(\mathbf{w}_t)\|^2 \left(1 - \frac{\eta(\lambda + \|\nabla^2 \mathcal{L}(\mathbf{v})\|)}{2}\right) \leq \widetilde{\mathcal{L}}(\mathbf{w}_t) - \frac{\eta}{2}\|\nabla \widetilde{\mathcal{L}}(\mathbf{w}_t)\|^2.$$

The risk contraction follows from the above and the well-known Polyak-Lojasiewicz inequality from the $\lambda$-strong convexity:

$$\widetilde{\mathcal{L}}(\mathbf{w}) - \min \widetilde{\mathcal{L}} \leq \frac{1}{2\lambda}\|\nabla \widetilde{\mathcal{L}}(\mathbf{w})\|^2.$$

The norm contraction is because

$$\|\mathbf{w}_{t+1} - \mathbf{w}_\lambda\|^2$$
$$= \|\mathbf{w}_t - \mathbf{w}_\lambda\|^2 + 2\eta \langle \nabla \widetilde{\mathcal{L}}(\mathbf{w}_t), \mathbf{w}_\lambda - \mathbf{w}_t \rangle + \eta^2 \|\nabla \widetilde{\mathcal{L}}(\mathbf{w}_t)\|^2$$
$$\leq \|\mathbf{w}_t - \mathbf{w}_\lambda\|^2 + 2\eta \left(\widetilde{\mathcal{L}}(\mathbf{w}_\lambda) - \widetilde{\mathcal{L}}(\mathbf{w}_t) - \frac{\lambda}{2}\|\mathbf{w}_t - \mathbf{w}_\lambda\|^2\right) + \eta^2 \left(\frac{2}{\eta}\left(\widetilde{\mathcal{L}}(\mathbf{w}_t) - \widetilde{\mathcal{L}}(\mathbf{w}_{t+1})\right)\right)$$
$$= (1 - \eta\lambda)\|\mathbf{w}_t - \mathbf{w}_\lambda\|^2 + 2\eta\left(\widetilde{\mathcal{L}}(\mathbf{w}_\lambda) - \widetilde{\mathcal{L}}(\mathbf{w}_{t+1})\right)$$
$$\leq (1 - \eta\lambda)\|\mathbf{w}_t - \mathbf{w}_\lambda\|^2,$$

where the first inequality is by $\lambda$-strong convexity and the first claim, and the second inequality is because $\mathbf{w}_\lambda := \arg\min \widetilde{\mathcal{L}}(\cdot)$. This completes our proof. $\qquad\square$

## A.2    Proof of Theorem 1

The following lemma is crucial for showing that GD remains in the stable phase.

**Lemma 13** (Stable phase). *Assume that $n \geq 2$ and*

$$\lambda \leq \frac{\gamma^2}{C_1} \min\left\{\frac{1}{n\ln n}, \frac{1}{n\eta}, \frac{1}{\eta^2}\right\}$$

*for a large constant $C_1 > 1$. If in the $s$-th step we have*

$$\mathcal{L}(\mathbf{w}_s) \leq \min\left\{\frac{1}{2e^2\eta}, \frac{\ln 2}{n}\right\},$$

*then for all $t \geq s$ we have*

$$\mathcal{L}(\mathbf{w}_t) \leq \min\left\{\frac{1}{2e^2\eta}, \frac{\ln 2}{n}\right\}.$$

*Proof of Lemma 13.* The condition on $\lambda$ with $C_1 \geq 6$ guarantees that

$$\lambda \leq \frac{1}{6}\min\left\{\frac{1}{n\ln n}, \frac{1}{\eta^2}\right\} \leq \frac{1}{6}\min\left\{1, \frac{1}{\eta^2}\right\} \leq \frac{1}{3(1 + \eta^2)} \leq \frac{1}{3\eta\ln(e + \eta)} \leq \frac{1}{2\eta}.$$

which satisfies the condition on $\lambda$ required by Lemmas 8, 11 and 12.

We prove the claim by induction. The claim holds for $s$. Assume the claim holds for $t$. We then verify the claim for $t + 1$. By Taylor's theorem, there exists $\mathbf{v}$ within the line segment between $\mathbf{w}_t$ and $\mathbf{w}_{t+1}$ such that

$$\mathcal{L}(\mathbf{w}_{t+1}) - \mathcal{L}(\mathbf{w}_t) = \langle \nabla \mathcal{L}(\mathbf{w}_t), \mathbf{w}_{t+1} - \mathbf{w}_t \rangle + \frac{1}{2}\langle \nabla^2 \mathcal{L}(\mathbf{v}), (\mathbf{w}_{t+1} - \mathbf{w}_t)^{\otimes 2} \rangle$$

$$= -\eta\langle\nabla\mathcal{L}(\mathbf{w}_t), \nabla\mathcal{L}(\mathbf{w}_t) + \lambda\mathbf{w}_t\rangle + \frac{\eta^2}{2}\langle\nabla^2\mathcal{L}(\mathbf{v}), (\nabla\mathcal{L}(\mathbf{w}_t) + \lambda\mathbf{w}_t)^{\otimes 2}\rangle$$

$$\leq -\eta\langle\nabla\mathcal{L}(\mathbf{w}_t), \nabla\mathcal{L}(\mathbf{w}_t) + \lambda\mathbf{w}_t\rangle + \frac{\eta^2}{2}\mathcal{G}(\mathbf{v})\|\nabla\mathcal{L}(\mathbf{w}_t) + \lambda\mathbf{w}_t\|^2.$$

where the last inequality is because $\|\nabla^2\mathcal{L}(\mathbf{v})\| \leq \mathcal{G}(\mathbf{v})$ by Lemma 4. The induction hypothesis and Lemma 4 imply that

$$\mathcal{G}(\mathbf{w}_t) \leq \mathcal{L}(\mathbf{w}_t) \leq \min\left\{\frac{1}{2e^2\eta}, \frac{\ln 2}{n}\right\}, \tag{3}$$

which implies $\mathcal{G}(\mathbf{v}) \leq 2/\eta \leq 1/\eta$ by Lemma 11. Then we have

$$\mathcal{L}(\mathbf{w}_{t+1}) - \mathcal{L}(\mathbf{w}_t) \leq -\eta\langle\nabla\mathcal{L}(\mathbf{w}_t), \nabla\mathcal{L}(\mathbf{w}_t) + \lambda\mathbf{w}_t\rangle + \frac{\eta}{2}\|\nabla\mathcal{L}(\mathbf{w}_t) + \lambda\mathbf{w}_t\|^2$$

$$= -\frac{\eta}{2}\Big(\|\nabla\mathcal{L}(\mathbf{w}_t)\|^2 - \lambda\|\mathbf{w}_t\|^2\Big).$$

We discuss two cases. If $\|\nabla\mathcal{L}(\mathbf{w}_t)\| \geq \lambda\|\mathbf{w}_t\|$, then $\mathcal{L}(\mathbf{w}_{t+1}) \leq \mathcal{L}(\mathbf{w}_t)$, which together with (3) verifies the claim for $t + 1$. If $\|\nabla\mathcal{L}(\mathbf{w}_t)\| < \lambda\|\mathbf{w}_t\|$, then

$$\mathcal{L}(\mathbf{w}_{t+1}) \leq \mathcal{L}(\mathbf{w}_t) + \frac{\eta\lambda^2\|\mathbf{w}_t\|^2}{2}$$

$$\leq 2\mathcal{G}(\mathbf{w}_t) + \frac{\eta\lambda^2\|\mathbf{w}_t\|^2}{2}$$

$$\leq \frac{2}{\gamma}\|\nabla\mathcal{L}(\mathbf{w}_t)\| + \frac{\eta\lambda^2\|\mathbf{w}_t\|^2}{2}$$

$$\leq \frac{\lambda\|\mathbf{w}_t\|}{\gamma}\left(2 + \frac{\gamma\eta\lambda\|\mathbf{w}_t\|}{2}\right),$$

where the second inequality is by Lemma 4 and (3), the third inequality is by Lemma 4, and the fourth inequality is because $\|\nabla\mathcal{L}(\mathbf{w}_t)\| < \lambda\|\mathbf{w}_t\|$. By Lemma 8, we have

$$\|\mathbf{w}_t\| \leq 4\frac{\eta + \ln(e + \gamma^2\min\{\eta t, 1/\lambda\})}{\gamma} \leq 4\frac{\eta + \ln(e + \gamma^2/\lambda)}{\gamma} =: M.$$

Then we have

$$\mathcal{L}(\mathbf{w}_{t+1}) \leq \frac{\lambda M}{\gamma}\left(2 + \frac{\gamma\eta\lambda M}{2}\right) \leq 3\frac{\lambda M}{\gamma} \leq \min\left\{\frac{1}{2e^2\eta}, \frac{\ln 2}{n}\right\},$$

which verifies the claim for $t + 1$. Here, the last inequality is by (4) (proved below), and the second inequality is because $\eta\lambda M \leq 1$ by (4).

$$\frac{3\lambda M}{\gamma} = 12\lambda\frac{\eta + \ln(e + \gamma^2/\lambda)}{\gamma^2} \leq \min\left\{\frac{1}{2e^2\eta}, \frac{\ln 2}{n}\right\} \tag{4}$$

$$\Leftarrow \quad \frac{\eta + \ln(e + \gamma^2/\lambda)}{\gamma^2/\lambda} \leq \frac{1}{K_1}\min\left\{\frac{1}{\eta}, \frac{1}{n}\right\} \quad \text{for a sufficiently large constant } K_1 \tag{5}$$

$$\Leftrightarrow \quad \frac{\gamma^2}{\lambda} \geq K_1\big(\eta^2 + \eta\ln(e + \gamma^2/\lambda)\big) \quad\text{and}\quad \frac{\gamma^2}{\lambda} \geq K_1\big(n\eta + n\ln(e + \gamma^2/\lambda)\big)$$

$$\Leftarrow \quad \frac{\gamma^2}{\lambda} \geq C_1\max\{1, \eta^2\} \text{ and } \frac{\gamma^2}{\lambda} \geq C_1\max\{n\eta, n\ln n\} \text{ for a sufficiently large constant } C_1$$

$$\Leftrightarrow \quad \frac{\gamma^2}{\lambda} \geq C_1\max\{n\ln n, n\eta, \eta^2\}.$$

This completes the proof. □

The next lemma provides a bound on the phase transition time.

**Lemma 14** (Phase transition). *Assume that $n \geq 2$ and*

$$\lambda \leq \frac{\gamma^2}{C_1}\min\left\{\frac{1}{n\ln n}, \frac{1}{n\eta}, \frac{1}{\eta^2}\right\}$$

*for a large constant $C_1 > 1$. Let*

$$\tau := \frac{C_2}{\gamma^2} \max\left\{\eta,\, n,\, \frac{n \ln n}{\eta}\right\}$$

*for a large constant $C_2 > 1$. Then for all $t \geq \tau$,*

$$\mathcal{G}(\mathbf{w}_t) \leq \mathcal{L}(\mathbf{w}_t) \leq \min\left\{\frac{1}{2e^2\eta},\, \frac{\ln 2}{n}\right\}.$$

*Proof of Lemma 14.* The assumption on $\lambda$ with $C_1 \geq 2$ implies $\eta\lambda \leq 1/2$. Then by Lemma 9 we have

$$\frac{1}{\tau}\sum_{k=0}^{\tau-1} \mathcal{G}(\mathbf{w}_k) \leq 11 \frac{\eta + \ln(e + \gamma^2\min\{\eta\tau, 1/\lambda\})}{\gamma^2\min\{\eta\tau, 1/\lambda\}}.$$

If the right-hand side is smaller than $\min\{1/(4e^2\eta), \ln(2)/(2n)\}$, then there exists $s \leq \tau$ such that

$$\mathcal{G}(\mathbf{w}_s) \leq \min\left\{\frac{1}{4e^2\eta},\, \frac{\ln 2}{2n}\right\} \leq \frac{1}{2n}.$$

This further implies $\mathcal{L}(\mathbf{w}_s) \leq 2\mathcal{G}(\mathbf{w}_s) \leq \min\{1/(2e^2\eta), \ln(2)/n\}$ by Lemma 4, and then Lemmas 4 and 13 imply the result. So it suffices to check that

$$11\frac{\eta + \ln(e + \gamma^2\min\{\eta\tau, 1/\lambda\})}{\gamma^2\min\{\eta\tau, 1/\lambda\}} \leq \min\left\{\frac{1}{4e^2\eta},\, \frac{\ln 2}{2n}\right\}$$

$$\Leftarrow \quad \begin{cases} \dfrac{\eta + \ln(e + \gamma^2/\lambda)}{\gamma^2/\lambda} \leq \dfrac{1}{K_1}\min\left\{\dfrac{1}{\eta}, \dfrac{1}{n}\right\}, \\[2ex] \dfrac{\eta + \ln(e + \gamma^2\eta\tau)}{\gamma^2\eta\tau} \leq \dfrac{1}{K_1}\min\left\{\dfrac{1}{\eta}, \dfrac{1}{n}\right\}, \end{cases}$$

for a sufficiently large constant $K_1$. As shown in (5) in the proof of Lemma 13, the first condition follows from our assumption on $\lambda$. The second condition is equivalent to

$$\gamma^2\tau \geq K_1\big(\eta + \ln(e + \gamma^2\eta\tau)\big) \text{ and } \gamma^2\tau \geq K_1\left(n + \frac{n}{\eta}\ln(e + \gamma^2\eta\tau)\right)$$

$$\Leftarrow \quad \gamma^2\tau \geq C_2\max\{1, \eta\} \text{ and } \gamma^2\tau \geq C_2\max\left\{n, \frac{n\ln n}{\eta}\right\} \text{ for a sufficiently large constant } C_2$$

$$\Leftrightarrow \quad \gamma^2\tau \geq C_2\max\left\{\eta,\, n,\, \frac{n\ln n}{\eta}\right\}.$$

This completes the proof. $\qquad\square$

With the above lemmas, we are ready to prove Theorem 1.

*Proof of Theorem 1.* Our assumption on $\lambda$ and $\eta$ satisfies the condition on $\lambda$ and $\eta$ required by Lemma 14. That condition with $C_1 \geq 6$ implies that $\lambda \leq 1/(3(1+\eta^2)) \leq 1/(3\eta\ln(e+\eta)) \leq 1/(2\eta)$, satisfying the condition on $\lambda$ and $\eta$ required by Lemmas 8 and 12.

The phase transition time bound is by Lemma 14, which further enables Lemma 12 for all $t \geq \tau$. Thus we have

$$\widetilde{\mathcal{L}}(\mathbf{w}_{\tau+t}) - \min\widetilde{\mathcal{L}} \leq (1-\lambda\eta)^t\big(\widetilde{\mathcal{L}}(\mathbf{w}_\tau) - \min\widetilde{\mathcal{L}}\big) \leq \exp(-\lambda\eta t)\big(\widetilde{\mathcal{L}}(\mathbf{w}_\tau) - \min\widetilde{\mathcal{L}}\big),$$

$$\|\mathbf{w}_{\tau+t} - \mathbf{w}_\lambda\|^2 \leq (1-\lambda\eta)^t\|\mathbf{w}_\tau - \mathbf{w}_\lambda\|^2 \leq \exp(-\lambda\eta t)\|\mathbf{w}_\tau - \mathbf{w}_\lambda\|^2.$$

It remains to bound $\widetilde{\mathcal{L}}(\mathbf{w}_\tau) - \min\widetilde{\mathcal{L}}$ and $\|\mathbf{w}_\tau - \mathbf{w}_\lambda\|$.

Our assumption on $\lambda$ implies $\gamma^2/\lambda \geq C_1 \geq e$. Then by Lemma 8 we have

$$\|\mathbf{w}_\tau\| \leq 4\frac{\eta + \ln(e + \gamma^2\min\{\eta s, 1/\lambda\})}{\gamma} \leq 4\frac{\eta + \ln 2 + \ln(\gamma^2/\lambda)}{\gamma} \leq 4\frac{\eta + 2\ln(\gamma^2/\lambda)}{\gamma}.$$

We then use Lemma 3 to bound $\|\mathbf{w}_\tau - \mathbf{w}_\lambda\|$ by

$$\|\mathbf{w}_\tau - \mathbf{w}_\lambda\| \leq \|\mathbf{w}_\tau\| + \|\mathbf{w}_\lambda\| \leq 4\frac{\eta + 2\ln(\gamma^2/\lambda)}{\gamma} + \frac{\sqrt{2} + \ln(\gamma^2/\lambda)}{\gamma} \leq 10\frac{\eta + \ln(\gamma^2/\lambda)}{\gamma}.$$

This completes our proof for the parameter convergence.

Furthermore, the assumption on $\lambda$ guarantees that

$$\frac{\lambda}{\gamma^2} \lesssim 1, \quad \frac{\lambda}{\gamma^2}\eta^2 \lesssim 1, \quad \frac{\lambda}{\gamma^2}\ln^2(\gamma^2/\lambda) \lesssim 1.$$

Therefore, we have

$$\frac{\lambda}{2}\|\mathbf{w}_\tau\|^2 \leq 8\frac{\lambda}{\gamma^2}\left(\eta + 2\ln(\gamma^2/\lambda\})\right)^2 \leq C_3 - 1$$

for a constant $C_3 > 1$. Also note that $\mathcal{L}(\mathbf{w}_\tau) \leq \min\{1/(2e^2\eta), \ln(2)/n\} \leq 1$ by Lemma 14. These two bounds together imply that

$$\widetilde{\mathcal{L}}(\mathbf{w}_\tau) - \min\widetilde{\mathcal{L}} \leq \widetilde{\mathcal{L}}(\mathbf{w}_\tau) = \mathcal{L}(\mathbf{w}_\tau) + \frac{\lambda}{2}\|\mathbf{w}_\tau\|^2 \leq C_3.$$

This completes our proof for the risk convergence. $\qquad\square$

## A.3  Proof of Corollary 2

*Proof of Corollary 2.* By Theorem 1, GD enters the stable phase in $\tau$ steps, and then attains an $\varepsilon$ error within an additional

$$t - \tau \lesssim \frac{\ln(1/\varepsilon)}{\eta\lambda} \approx \max\left\{\frac{1}{\gamma\sqrt{\lambda}}, \frac{n}{\gamma^2}\right\}\ln(1/\varepsilon)$$

steps. We can further upper bound the phase transition time by

$$\tau \approx \frac{1}{\gamma^2}\max\left\{\eta,\, n,\, \frac{n}{\eta}\ln\frac{n}{\eta}\right\}$$

$$\approx \frac{1}{\gamma^2}\max\{\eta,\, n\}$$

$$\approx \frac{1}{\gamma^2}\max\left\{\min\left\{\frac{\gamma}{\sqrt{\lambda}}, \frac{\gamma^2}{n\lambda}\right\},\, n\right\}$$

$$\approx \frac{1}{\gamma^2}\max\left\{\frac{\gamma}{\sqrt{\lambda}},\, n\right\},$$

where the first equality is by the definition of $\tau$, the second equality is because

$$\eta \approx \min\left\{\sqrt{\frac{\gamma^2}{\lambda}}, \frac{\gamma^2}{\lambda n}\right\} \gtrsim \min\left\{\sqrt{n\ln n}, \ln n\right\} \gtrsim \ln n,$$

the third equality is by the choice of $\eta$, and the fourth equality is because $\max\{\min\{a, a^2/b\}, b\} = \max\{a, b\}$ for $a, b > 0$. So the total number of steps is $t \lesssim \max\{1/(\gamma\sqrt{\lambda}, n/\gamma^2)\}\ln(1/\varepsilon)$. $\qquad\square$

## A.4  Proof of Theorem 4

The following lemma shows that GD stays in the stable phase.

**Lemma 15** (Stable phase, version 2). *Assume that*

$$\lambda \leq \frac{\gamma^2}{C_1}\min\left\{1, \frac{1}{\eta^3}\right\}$$

*for a large constant $C_1$. If in the s-th step we have*

$$\mathcal{L}(\mathbf{w}_s) \leq \frac{1}{4e^2\eta},$$

*then for all $t \geq s$ we have*

$$\mathcal{L}(\mathbf{w}_t) \leq \frac{1}{2e^2\eta}.$$

*Proof of Lemma 15.* The condition on $\lambda$ with $C_1 \geq 2$ implies $\lambda\eta \leq \min\{\eta, 1/\eta^2\}/2 \leq 1/2$. Then by Lemma 8 we have

$$\text{for all } t, \quad \|\mathbf{w}_t\| \leq 4\frac{\eta + \ln(e + \gamma^2 \min\{\eta s, 1/\lambda\})}{\gamma} \leq 4\frac{\eta + \ln(e + \gamma^2/\lambda)}{\gamma} =: M.$$

Our assumption on $\lambda$ implies that

$$\frac{\lambda}{2}M^2 = \frac{8\lambda\big(\eta + \ln(e + \gamma^2/\lambda)\big)^2}{\gamma^2} \leq \frac{1}{4e^2\eta}.$$

To see this, it is sufficient to check that

$$\frac{\lambda}{\gamma^2}(\eta^2 + \ln^2(e + \gamma^2/\lambda)) \leq \frac{1}{K_1\eta} \quad \text{for a sufficiently large constant } K_1 \tag{6}$$

$$\Leftarrow \quad \frac{\lambda}{\gamma^2} \leq \frac{1}{K_2\eta^3} \quad \text{and} \quad \frac{\lambda}{\gamma^2}\ln^2(e + \gamma^2/\lambda)) \leq \frac{1}{K_2\eta} \quad \text{for a sufficiently large constant } K_2$$

$$\Leftarrow \quad \frac{\lambda}{\gamma^2} \leq \frac{1}{C_1}\min\left\{1, \frac{1}{\eta^3}\right\} \quad \text{for a sufficiently large constant } C_1.$$

With this, we prove the following stronger claim by induction:

$$\text{for all } t \geq s, \quad \mathcal{L}(\mathbf{w}_t) \leq \widetilde{\mathcal{L}}(\mathbf{w}_t) \leq \frac{1}{2e^2\eta}.$$

In the $s$-th step, we have

$$\mathcal{L}(\mathbf{w}_s) \leq \widetilde{\mathcal{L}}(\mathbf{w}_s) \leq \mathcal{L}(\mathbf{w}_s) + \frac{\lambda}{2}\|\mathbf{w}_s\|^2 \leq \mathcal{L}(\mathbf{w}_s) + \frac{\lambda}{2}M^2 \leq \frac{1}{2e^2\eta},$$

which satisfies the hypothesis. Next, assume the hypothesis holds for $t$. Then $\mathcal{G}(\mathbf{w}_t) \leq \mathcal{L}(\mathbf{w}_t) \leq 1/(2e^2\eta)$. Additionally, our assumption on $\lambda$ with $C_1 \geq 6$ implies $\lambda \leq 1/(3(1+\eta^3)) \leq 1/(3\eta\ln(e+\eta))$. Thus we can apply Lemma 12 for $t$, obtaining that $\widetilde{\mathcal{L}}(\mathbf{w}_{t+1}) \leq \widetilde{\mathcal{L}}(\mathbf{w}_t) \leq 1/(2e^2\eta)$. This verifies the hypothesis for $t + 1$, and completes our induction. $\qquad\square$

The next lemma provides a bound on the phase transition time.

**Lemma 16** (Phase transition, version 2). *Assume that*

$$\lambda \leq \frac{\gamma^2}{C_1}\min\left\{1, \frac{1}{\eta^3}\right\}$$

*for a constant $C_1 > 1$. Let*

$$\tau := \frac{C_2\max\{1, \eta^2\}}{\gamma^2}$$

*for a constant $C_2 > 1$. Then for all $t \geq \tau$, we have*

$$\mathcal{G}(\mathbf{w}_t) \leq \mathcal{L}(\mathbf{w}_t) \leq \frac{1}{2e^2\eta}.$$

*Proof of Lemma 16.* The condition on $\lambda$ with $C_1 \geq 2$ implies $\lambda\eta \leq \min\{\eta, 1/\eta^2\}/2 \leq 1/2$. Then by Lemma 10 we have

$$\frac{1}{\tau}\sum_{k=0}^{\tau-1}\mathcal{L}(\mathbf{w}_k) \leq 10\frac{\eta^2 + \ln^2(e + \gamma^2\min\{\eta\tau, 1/\lambda\})}{\gamma^2\min\{\eta\tau, 1/\lambda\}}.$$

If the right-hand side is smaller than $1/(4e^2\eta)$, then there exists $s \leq \tau$ such that $\mathcal{L}(\mathbf{w}_s) \leq 1/(4e^2\eta)$. By Lemma 15, we have $\mathcal{L}(\mathbf{w}_t) \leq 1/(2e^2\eta)$ for all $t \geq s$. We then complete the proof by using $\mathcal{G}(\mathbf{w}) \leq \mathcal{L}(\mathbf{w})$ from Lemma 4.

To see the right-hand side is smaller than $1/(4e^2\eta)$, it suffices to show that

$$\frac{\eta^2 + \ln^2(e + \gamma^2/\lambda))}{\gamma^2/\lambda} \leq \frac{1}{K_1\eta} \quad \text{and} \quad \frac{\eta^2 + \ln^2(e + \gamma^2\eta\tau)}{\gamma^2\eta\tau} \leq \frac{1}{K_1\eta}$$

for a sufficiently large constant $K_1$. This first condition is implied by our assumption on $\lambda$ as shown by (6) in the proof of Lemma 15. For the second condition to hold, it suffices to have

$$\gamma^2\tau \geq K_2\eta^2 \quad \text{and} \quad \gamma^2\tau \geq K_2\ln^2(e+\eta\gamma^2\tau) \quad \text{for a sufficiently large constant } K_2$$
$$\Leftarrow \quad \gamma^2\tau \geq C_2\max\{1,\,\eta^2\} \quad \text{for a sufficiently large constant } C_2.$$

This completes the proof. $\qquad\square$

The proof of Theorem 4 follows from the above lemmas.

*Proof of Theorem 4.* Our assumption on $\lambda$ and $\eta$ satisfies the condition on $\lambda$ and $\eta$ required by Lemma 16. That condition with $C_1 \geq 6$ implies that $\lambda \leq 1/(3(1+\eta^3)) \leq 1/(3\eta\ln(e+\eta))$, satisfying the condition on $\lambda$ and $\eta$ required by Lemma 12.

The phase transition time bound is by Lemma 16, which further enables Lemma 12 for all $t \geq \tau$. Thus we have

$$\widetilde{\mathcal{L}}(\mathbf{w}_{\tau+t}) - \min\widetilde{\mathcal{L}} \leq (1-\lambda\eta)^t\big(\widetilde{\mathcal{L}}(\mathbf{w}_\tau) - \min\widetilde{\mathcal{L}}\big) \leq \exp(-\lambda\eta t)\big(\widetilde{\mathcal{L}}(\mathbf{w}_\tau) - \min\widetilde{\mathcal{L}}\big),$$
$$\|\mathbf{w}_{\tau+t} - \mathbf{w}_\lambda\|^2 \leq (1-\lambda\eta)^t\|\mathbf{w}_\tau - \mathbf{w}_\lambda\|^2 \leq \exp(-\lambda\eta t)\|\mathbf{w}_\tau - \mathbf{w}_\lambda\|^2.$$

Moreover, from the proof of Lemma 15, we know

$$\widetilde{\mathcal{L}}(\mathbf{w}_s) - \min\widetilde{\mathcal{L}} \leq \widetilde{\mathcal{L}}(\mathbf{w}_s) \leq \frac{1}{2e^2\eta}.$$

This completes our proof for the risk convergence. We need to bound $\|\mathbf{w}_s - \mathbf{w}_\lambda\|$ to complete our proof for the parameter convergence, which follows from the same argument as in the proof of Theorem 1 in Appendix A.2. $\qquad\square$

### A.5 Proof of Corollary 5

*Proof of Corollary 5.* Recall that $\eta \asymp \gamma^{2/3}/\lambda^{1/3}$. By Theorem 4, GD enters the stable phase in $\tau$ steps, and then attains an $\varepsilon$-suboptimal error within an additional

$$t - \tau \lesssim \frac{\ln(1/(\eta\varepsilon))}{\eta\lambda} \lesssim \frac{\ln(1/\varepsilon)}{(\gamma\lambda)^{2/3}}$$

steps. We can further upper bound the phase transition time by

$$\tau \asymp \frac{\max\{1,\,\eta^2\}}{\gamma^2} \asymp \frac{1}{(\gamma\lambda)^{2/3}}.$$

So the total number of steps is $t \lesssim \ln(1/\varepsilon)/(\gamma\lambda)^{2/3}$. $\qquad\square$

## B  A lower bound

The next lemma provides a hard dataset for which GD cannot use a large stepsize if it operates in the stable regime. The hard dataset construction is motivated by the lower bound of Wu et al. (2024).

**Lemma 17** (A stepsize bound). *Consider the dataset*

$$\mathbf{x}_1 = (\gamma, 0.9), \quad \mathbf{x}_2 = (\gamma, -0.5), \quad y_1 = y_2 = 1, \quad 0 < \gamma < 0.1.$$

*Then with $\mathbf{w}_0 = 0$, $\widetilde{\mathcal{L}}(\mathbf{w}_1) \leq \widetilde{\mathcal{L}}(\mathbf{w}_0)$ implies that $\eta \leq 20$.*

*Proof of Lemma 17.* We have $\widetilde{\mathcal{L}}(\mathbf{w}_0) = \ln 2$ and

$$\nabla\widetilde{\mathcal{L}}(\mathbf{w}_0) = -\frac{1}{2}(\gamma, 0.4) \quad \Rightarrow \quad \mathbf{w}_1 = \mathbf{w}_0 - \eta\nabla\widetilde{\mathcal{L}}(\mathbf{w}_0) = \frac{\eta}{2}(\gamma, 0.4).$$

So we have

$$\widetilde{\mathcal{L}}(\mathbf{w}_1) \geq \mathcal{L}(\mathbf{w}_1) \geq \frac{1}{2}\ln(1+\exp(-\mathbf{x}_2^\top\mathbf{w}_1)) = \frac{1}{2}\ln(1+\exp((\gamma^2+0.2)\eta/2)) \geq \frac{(\gamma^2+0.2)\eta}{4}.$$

Thus $\widetilde{\mathcal{L}}(\mathbf{w}_1) \leq \widetilde{\mathcal{L}}(\mathbf{w}_0)$ implies that $\eta \leq 4\ln(2)/(\gamma^2+0.2) \leq 20$, which completes the proof. $\qquad\square$

The following lemma establishes upper and lower bounds for the logistic empirical risk. For simplicity, this lemma is stated for the special dataset in Lemma 17. However, this lemma can be extended to general datasets satisfying Assumptions 1 and 3 using techniques from Wu et al. (2023).

**Lemma 18** (Upper and lower bounds on the logistic empirical risk). *Assume that $\lambda\eta < 1$. For the dataset in Lemma 17, we have*

$$\frac{1}{Ct} \leq \mathcal{L}(\mathbf{w}_t) \leq \frac{C}{t} \ \ and \ \ \|\mathbf{w}_t\| \leq C\ln(t), \ \ for \ 1 \leq t \leq \frac{1}{C\lambda\ln(1/\lambda)},$$

*where $C > 1$ depends on $\gamma$ and $\eta$ but is independent of $t$ and $\lambda$.*

*Proof of Lemma 18.* Denote the trainable parameter as $\mathbf{w} = (w, \bar{w})$. Then we have $w_0 = \bar{w}_0 = 0$ and

$$w_{t+1} = (1 - \eta\lambda)w_t + \frac{\eta\gamma}{2}\left(\frac{1}{1 + e^{\gamma w_t + 0.9\bar{w}_t}} + \frac{1}{1 + e^{\gamma w_t - 0.5\bar{w}_t}}\right) \tag{7}$$

$$\bar{w}_{t+1} = (1 - \eta\lambda)\bar{w}_t + \frac{\eta}{2}\left(\frac{0.9}{1 + e^{\gamma w_t + 0.9\bar{w}_t}} - \frac{0.5}{1 + e^{\gamma w_t - 0.5\bar{w}_t}}\right). \tag{8}$$

**Bounds on $\bar{w}_t$.** Recall that $\eta\lambda < 1$. From (7), we see that $(w_t)_{t\geq 0}$ are all nonnegative. Then by direct computation, we can verify that the factor within the big bracket in (8) is positive when $\bar{w}_t \leq 0$ and is negative when $\bar{w}_t \geq 2$. Then (8) implies the following:

$$\begin{array}{lll} \text{if } \bar{w}_t \leq 0, & -(1 - \eta\lambda)|\bar{w}_t| \leq (1 - \eta\lambda)\bar{w}_t \leq \bar{w}_{t+1} \leq (1 - \eta\lambda)\bar{w}_t + \eta \leq \eta; \\ \text{if } 0 < \bar{w}_t \leq 2, & -\eta \leq (1 - \eta\lambda)\bar{w}_t - \eta \leq \bar{w}_{t+1} \leq (1 - \eta\lambda)\bar{w}_t + \eta \leq 2 + \eta; \\ \text{if } \bar{w}_t > 2, & -\eta \leq (1 - \eta\lambda)\bar{w}_t - \eta \leq \bar{w}_{t+1} \leq (1 - \eta\lambda)\bar{w}_t \leq (1 - \eta\lambda)|\bar{w}_t|. \end{array}$$

In all cases, we have

$$|\bar{w}_{t+1}| \leq \max\{(1 - \eta\lambda)|\bar{w}_t|, \eta + 2\},$$

which implies that $|\bar{w}_t| \leq \eta + 2$ for every $t \geq 0$ by induction.

Let

$$\mathcal{H}(\bar{w}) := \frac{1}{2}\big(\exp(-0.9\bar{w}) + \exp(0.5\bar{w})\big).$$

Then for every $t \geq 0$, we have $\mathcal{H}(\bar{w}_t) \leq \exp(\eta + 2) := H_{\max}$.

**An upper bound on $w_t$.** Using the upper bound on $\mathcal{H}(\bar{w}_t)$ and (7), we have

$$w_{t+1} \leq w_t + \frac{\eta\gamma}{2}\left(\frac{1}{e^{\gamma w_t + 0.9\bar{w}_t}} + \frac{1}{e^{\gamma w_t - 0.5\bar{w}_t}}\right) \leq w_t + \frac{\eta\gamma H_{\max}}{2}e^{-\gamma w_t}.$$

Let $t_0 := \min\{t : \gamma^2\eta H_{\max}\exp(-\gamma w_t)/2 \leq 1\}$. Since $w_t$ is increasing, $t_0$ exists. For every $t \leq t_0$, we have $w_t \leq \gamma^{-1}\ln(\gamma^2\eta H_{\max}/2)$. For $t \geq t_0$, we have

$$e^{\gamma w_{t+1}} \leq e^{\gamma w_t}e^{\gamma^2\eta H_{\max}/2\exp(-\gamma w_t)} \leq e^{\gamma w_t}\left(1 + e^{\frac{\gamma^2\eta H_{\max}\exp(-\gamma w_t)}{2}}\right) \leq e^{\gamma w_t} + \frac{e\gamma^2\eta H_{\max}}{2}$$

$$\Rightarrow \quad \text{for } t \geq t_0, \quad w_t \leq \frac{1}{\gamma}\ln\left(\frac{e\gamma^2\eta H_{\max}}{2}(t - t_0) + e^{\gamma w_{t_0}}\right).$$

Putting these two bounds together, we have for every $t \geq 0$,

$$w_t \leq \frac{1}{\gamma}\ln\left(e\gamma^2\eta H_{\max}(t + 1)\right).$$

**A lower bound on $w_t$.** From (7), we have

$$w_{t+1} \geq (1 - \eta\lambda)w_t + \frac{\eta\gamma}{2}\big(\min\{1, e^{-\gamma w_t - 0.9\bar{w}_t}\} + \min\{1, e^{-\gamma w_t + 0.5\bar{w}_t}\}\big)$$

$$= (1 - \eta\lambda)w_t + \frac{\eta\gamma}{2}e^{-\gamma w_t}\big(\min\{e^{\gamma w_t}, e^{-0.9\bar{w}_t}\} + \min\{e^{\gamma w_t}, e^{0.5\bar{w}_t}\}\big)$$

$$\geq (1-\eta\lambda)w_t + \frac{\eta\gamma}{2}e^{-\gamma w_t}\big(\min\{1, e^{-0.9\bar{w}_t}\} + \min\{1, e^{0.5\bar{w}_t}\}\big)$$
$$\geq (1-\eta\lambda)w_t + \frac{\eta\gamma}{2}e^{-\gamma w_t},$$

For $t$ such that

$$\lambda < \frac{1}{4\eta H_{\max}(t+1)\ln\big(e\gamma^2\eta H_{\max}(t+1)\big)},$$

from our upper bound on $w_t$, we have

$$\frac{2\lambda}{\gamma}w_t e^{\gamma w_t} \leq \frac{1}{2}.$$

Then for such $t$, we have

$$e^{\gamma w_{t+1}} \geq e^{(1-\eta\lambda)\gamma w_t + \frac{\eta\gamma}{2}e^{-\gamma w_t}} = e^{\gamma w_t}\exp\left(\frac{\eta\gamma}{2}e^{-\gamma w_t}\left(1 - \frac{2\lambda}{\gamma}w_t e^{\gamma w_t}\right)\right)$$
$$\geq e^{\gamma w_t}\exp\left(\frac{\eta\gamma}{4}e^{-\gamma w_t}\right) \geq e^{\gamma w_t}\left(1 + \frac{\eta\gamma}{4}e^{-\gamma w_t}\right) \geq e^{\gamma w_t} + \frac{\eta\gamma}{4}.$$

So we have

$$w_t \geq \frac{1}{\gamma}\ln\left(\frac{\eta\gamma}{4}t + 1\right), \quad \text{for } t \lesssim \frac{1}{\gamma^2\eta H_{\max}\lambda\ln(1/\lambda)}.$$

**Bounds on logistic empirical risk.** Notice that

$$e^{-\gamma w_t + |\bar{w}_t|} \geq \mathcal{L}(\mathbf{w}_t) \geq \frac{1}{2}\ln(1 + e^{-\gamma w_t}).$$

This, together with our upper and lower bounds on $w_t$ and $\bar{w}_t$, leads to the promised bounds. □

With the above lemmas, we are ready to prove Theorem 3.

*Proof of Theorem 3.* From Lemma 17 we know $\eta \leq 20$. Then from Lemma 18, we have

$$\frac{1}{C_0 t} \leq \mathcal{L}(\mathbf{w}_t) \leq \frac{C_0}{t} \quad \text{and} \quad \|\mathbf{w}_t\| \leq C_0\ln(t), \quad \text{for } t \leq \frac{1}{C_0\lambda\ln(1/\lambda)},$$

where $C_0 > 1$ is a large factor that only depends on $\gamma$ but is independent of $t$ and $\lambda$. Here, we can make $C_0$ independent of $\eta$ as $\eta \leq 20$. For every sufficiently small $\lambda$, we can pick

$$\tau := \frac{1}{C_0^2\lambda\ln^2(1/\lambda)} < \frac{1}{C_0\lambda\ln(1/\lambda)}.$$

For this $\tau$, we have

$$C_0\lambda\ln^2(1/\lambda) \leq \mathcal{L}(\mathbf{w}_\tau) \leq C_0^3\lambda\ln^2(1/\lambda) \quad \text{and} \quad \|\mathbf{w}_\tau\| \leq C_0\ln(1/\lambda).$$

By Lemma 3 and setting $C_0$ large enough, we get

$$\min\widetilde{\mathcal{L}} \leq \frac{\lambda(2 + \ln^2(\gamma^2/\lambda))}{2\gamma^2} \leq \frac{1}{2}C_0\lambda\ln^2(1/\lambda).$$

That is, we have

$$\widetilde{\mathcal{L}}(\mathbf{w}_\tau) - \min\widetilde{\mathcal{L}} \geq \mathcal{L}(\mathbf{w}_\tau) - \min\widetilde{\mathcal{L}} \geq \frac{1}{2}C_0\lambda\ln^2(1/\lambda).$$

**Step complexity for a large $\varepsilon$.** For $\varepsilon \geq 0.5C_0\lambda\ln^2(1/\lambda)$, we have

$$\widetilde{\mathcal{L}}(\mathbf{w}_t) - \min\widetilde{\mathcal{L}} \leq \varepsilon \quad \Rightarrow \quad \mathcal{L}(\mathbf{w}_t) \leq \min\widetilde{\mathcal{L}} + \varepsilon \leq 2\varepsilon$$
$$\Rightarrow \quad t \geq \frac{1}{2C_0\varepsilon},$$

where the second line is because of the lower bound on $\mathcal{L}(\mathbf{w}_t)$ for $t \leq 1/(C_0\lambda\ln(1/\lambda))$ and our choice of $\lambda$.

**Step complexity for a small $\varepsilon$.** The case of $\varepsilon < 0.5C_0\lambda\ln^2(1/\lambda)$ needs some more effort. Since GD operates in the stable regime, we have

$$\text{for all } t \geq \tau, \quad \widetilde{\mathcal{L}}(\mathbf{w}_t) \leq \widetilde{\mathcal{L}}(\mathbf{w}_\tau) \leq \mathcal{L}(\mathbf{w}_\tau) + \frac{\lambda}{2}\|\mathbf{w}_\tau\|^2 \leq 2C_0^3\lambda\ln^2(1/\lambda).$$

That is, $(\mathbf{w}_t)_{t\geq\tau}$ are all within the level set

$$\mathcal{W} := \left\{\mathbf{w} : \widetilde{\mathcal{L}}(\mathbf{w}) \leq 2C_0^3\lambda\ln^2(1/\lambda)\right\}.$$

For parameters in this level set, we have

$$\sup_{\mathbf{w}\in\mathcal{W}} \|\nabla^2\widetilde{\mathcal{L}}(\mathbf{w})\| = \sup_{\mathbf{w}\in\mathcal{W}} \|\nabla^2\mathcal{L}(\mathbf{w})\| + \lambda$$

$$\begin{cases} \leq \sup_{\mathbf{w}\in\mathcal{W}} \mathcal{L}(\mathbf{w}) + \lambda \leq \sup_{\mathbf{w}\in\mathcal{W}} \widetilde{\mathcal{L}}(\mathbf{w}_t) + \lambda \leq 3C_0^3\lambda\ln^2(1/\lambda), \\ \geq \lambda, \end{cases}$$

where the upper bound is given by Lemma 4 and the definition of $\mathcal{W}$. That is, $\widetilde{\mathcal{L}}$ is $\beta$-smooth for $\beta = 3C_0^3\lambda\ln^2(1/\lambda)$ and $\lambda$-strongly convex for $\mathbf{w} \in \mathcal{W}$. By standard convex optimization theory, we have

$$\text{for all } \mathbf{w} \in \mathcal{W}, \quad \widetilde{\mathcal{L}}(\mathbf{w}) - \min\widetilde{\mathcal{L}} \geq \frac{1}{2\beta}\|\nabla\widetilde{\mathcal{L}}(\mathbf{w})\|^2;$$

$$\widetilde{\mathcal{L}}(\mathbf{w}_{t+1}) \geq \widetilde{\mathcal{L}}(\mathbf{w}_t) + \langle\nabla\widetilde{\mathcal{L}}(\mathbf{w}_t), \mathbf{w}_{t+1} - \mathbf{w}_t\rangle = \widetilde{\mathcal{L}}(\mathbf{w}_t) - \eta\|\nabla\widetilde{\mathcal{L}}(\mathbf{w}_t)\|^2.$$

Moreover, our choice of $\lambda$ implies $\eta \leq 20 < 1/(4\beta)$. Then the above two inequalities imply that

$$\text{for all } t \geq \tau, \quad \widetilde{\mathcal{L}}(\mathbf{w}_{t+1}) - \min\widetilde{\mathcal{L}} \geq (1 - 2\eta\beta)\big(\widetilde{\mathcal{L}}(\mathbf{w}_t) - \min\widetilde{\mathcal{L}}\big),$$

which further implies that

$$\widetilde{\mathcal{L}}(\mathbf{w}_t) - \min\widetilde{\mathcal{L}} \geq (1 - 2\eta\beta)^{t-\tau}\big(\widetilde{\mathcal{L}}(\mathbf{w}_\tau) - \min\widetilde{\mathcal{L}}\big)$$

$$\geq (1 - 2\eta\beta)^{t-\tau}\frac{1}{2}C_0\lambda\ln^2(1/\lambda)$$

$$\geq \exp\big(-4\eta\beta(t-\tau)\big)\frac{1}{2}C_0\lambda\ln^2(1/\lambda).$$

For the right-hand side to be smaller than $\varepsilon < 0.5C_0\lambda\ln^2(1/\lambda)$, we need

$$t \geq \tau + \frac{\ln\big(0.5C_0\lambda\ln^2(1/\lambda)/\varepsilon\big)}{4\eta\beta}$$

$$\geq \frac{1}{C_0\lambda\ln^2(1/\lambda)} + \frac{\ln\big(0.5C_0\lambda\ln^2(1/\lambda)/\varepsilon\big)}{240C_0^3\lambda\ln^2(1/\lambda)}.$$

This completes our proof. □

# C  Population risk analysis

We provide a proof for Proposition 6 in Appendix C.1, then calculate the optimal regularization hyperparameter in Appendix C.2.

## C.1  Proof of Proposition 6

*Proof of Proposition 6.* It is clear that under Assumption 2, $\|\mathbf{w}\| \leq B$ implies $\ell(y\mathbf{x}^\top\mathbf{w}) \leq \ell(0) + B$. Applying Srebro et al. (2010, Theorem 1) to the functional class induced by $\{\mathbf{w} : \|\mathbf{w}\| \leq B\}$, we have the following: with probability $1 - \delta$,

$$\text{for every } \mathbf{w} \text{ such that } \|\mathbf{w}\| \leq B, \quad \mathcal{L}_{\text{test}}(\mathbf{w}) \lesssim \mathcal{L}(\mathbf{w}) + \ln^3(n)\mathcal{R}_n^2(B) + \frac{(B+1)\ln(1/\delta)}{n},$$

where $\mathcal{R}_n(B)$ is the Rademacher complexity of the functional class induced by $\{\mathbf{w} : \|\mathbf{w}\| \leq B\}$,

$$\mathcal{R}_n(B) := \sup_{(\mathbf{x}_i, y_i)_{i=1}^n} \mathbb{E}_\sigma \sup_{\|\mathbf{w}\| \leq B} \frac{1}{n} \left| \sum_{i=1}^n \sigma_i \ell(-y_i \mathbf{x}_i^\top \mathbf{w}) \right|,$$

where $\sigma = (\sigma_i)_{i=}^n$ are $n$ independent Rademacher random varaibles.

We control the Rademacher complexity by (Shalev-Shwartz and Ben-David, 2014, Lemma 26.10)

$$\mathcal{R}_n(B) \leq \sup_{(\mathbf{x}_i, y_i)_{i=1}^n} \mathbb{E}_\sigma \sup_{\|\mathbf{w}\| \leq B} \frac{1}{n} \left| \sum_{i=1}^n \sigma_i y_i \mathbf{x}_i^\top \mathbf{w} \right| \leq \frac{B}{n} \sup_{(\mathbf{x}_i, y_i)_{i=1}^n} \mathbb{E}_\sigma \left\| \sum_{i=1}^n \sigma_i y_i \mathbf{x}_i \right\|$$

$$\leq \frac{B}{n} \sup_{(\mathbf{x}_i, y_i)_{i=1}^n} \sqrt{\mathbb{E}_\sigma \left\| \sum_{i=1}^n \sigma_i y_i \mathbf{x}_i \right\|^2} = \frac{B}{n} \sup_{(\mathbf{x}_i, y_i)_{i=1}^n} \sqrt{\sum_{i=1}^n \|\mathbf{x}_i\|^2} \leq \frac{B}{\sqrt{n}},$$

where the first inequality is by the 1-Lipschitzness of $\ell$, the second inequality is by Cauchy–Schwarz inequality, and the last inequality is by Assumption 2.

Putting these together, we have: with probability $1 - \delta$,

$$\text{for every } \mathbf{w} \text{ such that } \|\mathbf{w}\| \leq B, \quad \mathcal{L}_{\text{test}}(\mathbf{w}) \lesssim \mathcal{L}(\mathbf{w}) + \frac{B^2 \ln^3(n)}{n} + \frac{(B+1)\ln(1/\delta)}{n}.$$

Now for a given $\mathbf{w}$, consider a sequence of balls with radius $B_i = e^i$ and a sequence of probabilities $\delta_i = \delta/(i+1)^2$. It is clear that $\sum_i \delta_i \lesssim \delta$ and $\mathbf{w}$ belongs to $B_i$ for $i = \ln(\|\mathbf{w}\| + 1)$. Applying the above inequality to each $B_i$ and $\delta_i$, then applying a union bound (motivated by the proof of Theorem 26.14 in (Shalev-Shwartz and Ben-David, 2014)), we get: for every $\mathbf{w}$, with probability $1 - \delta$,

$$\mathcal{L}_{\text{test}}(\mathbf{w}) \lesssim \mathcal{L}(\mathbf{w}) + \frac{(\|\mathbf{w}\| + 1)^2 \ln^3(n)}{n} + \frac{(\|\mathbf{w}\| + 1)\ln(\ln(\|\mathbf{w}\| + 1)/\delta))}{n}$$

$$\lesssim \mathcal{L}(\mathbf{w}) + \frac{\max\{1, \|\mathbf{w}\|^2\}(\ln^3(n) + \ln(1/\delta))}{n}.$$

This completes the proof. $\qquad\square$

## C.2 Optimal regularization

We compute the optimal regularization hyperparameter $\lambda$ such that $\mathbf{w}_\lambda$ minimizes the upper bound in Proposition 6. We assume that $\lambda < 1/\gamma^2$. From Lemma 3, we have

$$\|\mathbf{w}_\lambda\| \leq \frac{\sqrt{2} + \ln(\gamma^2/\lambda)}{\gamma}, \quad \mathcal{L}(\mathbf{w}_\lambda) \leq \widetilde{\mathcal{L}}(\mathbf{w}_\lambda) \leq \frac{\lambda(2 + \ln^2(\gamma^2/\lambda))}{2\gamma^2}.$$

Pugging these into Proposition 6, we have

$$\mathcal{L}_{\text{test}}(\mathbf{w}_\lambda) \lesssim \mathcal{L}(\mathbf{w}_\lambda) + \frac{\ln^3(n) + \ln(1/\delta)}{n} \|\mathbf{w}_\lambda\|^2$$

$$\lesssim \frac{\lambda(1 + \ln^2(\gamma^2/\lambda))}{\gamma^2} + \frac{\ln^3(n) + \ln(1/\delta)}{n} \frac{1 + \ln^2(\gamma^2/\lambda)}{\gamma^2}.$$

Choosing $\lambda \asymp 1/n$ minimizes the right-hand side up to constant factors, where we have

$$\mathcal{L}_{\text{test}}(\mathbf{w}_\lambda) \lesssim \frac{(\ln^3(n) + \ln(1/\delta))\ln^2(n)}{\gamma^2 n} = \widetilde{\mathcal{O}}\left(\frac{1}{\gamma^2 n}\right).$$

Note that the upper bound provided in Proposition 6 is at least $\Omega(1/n)$. So the choice of $\lambda \asymp 1/n$ leads to a nearly unimprovable bound, ignoring logarithmic factors and dependence on $\gamma$.

## D The critical stepsize threshold

We first prove Theorem 7 in Appendix D.1. We then show that the proposed critical threshold also sharply determines the global convergence of GD in Appendix D.2.

## D.1 Proof of Theorem 7

Denote the linearly separable dataset in Assumption 1 as

$$\mathbf{X} := (\mathbf{x}_1, \ldots, \mathbf{x}_n)^\top, \quad \mathbf{y} := (y_1, \ldots, y_n)^\top.$$

Consider the following margin maximization program

$$\min_{\mathbf{w} \in \mathbb{H}} \ \|\mathbf{w}\| \quad \text{s.t.} \ \ y_i \mathbf{x}_i^\top \mathbf{w} \geq 1, \ i = 1, \ldots, n.$$

Its Lagrangian dual can be written

$$\max_{\boldsymbol{\beta} \in \mathbb{R}^n} \ -\frac{1}{2} \boldsymbol{\beta}^\top \mathbf{X} \mathbf{X}^\top \boldsymbol{\beta} + \boldsymbol{\beta}^\top \mathbf{y} \quad \text{s.t.} \ \ y_i \beta_i \geq 0, \ i = 1, \ldots, n,$$

where $y_i \beta_i$ is the dual variable associated with the $i$-th constraint (see, e.g., Hsu et al., 2021). Let $\hat{\boldsymbol{\beta}}$ be the solution to the above problem. Let

$$\mathcal{S}_+ := \{i \in [n] : y_i \hat{\beta}_i > 0\}$$

be the set of support vectors with nonzero dual variables. Then Assumption 3 says that $\{\mathbf{x}_i : i \in \mathcal{S}_+\}$ spans the same space as $\{\mathbf{x}_1, \ldots, \mathbf{x}_n\}$.

We introduce some additional notation following Wu et al. (2023). By the rotational invariance of the problem, we can assume without loss of generality that the maximum $\ell_2$-margin direction is aligned with the first vector of the canonical basis. Then we can write the dataset and the parameters as

$$\mathbf{x}_i = (x_i, \bar{\mathbf{x}}_i), \quad \mathbf{w} = (w, \bar{\mathbf{w}}), \quad \mathbf{w}_\lambda = (w_\lambda, \bar{\mathbf{w}}_\lambda),$$

where $y_i x_i \geq \gamma$ by Assumption 1.

Let

$$\mathcal{S} := \{i : y_i x_i = \gamma\}$$

be the index set of all support vectors (satisfying the constraint with equality). Then Assumption 3 implies that $(\bar{\mathbf{x}}_i, y_i)_{i \in \mathcal{S}}$ are strictly nonseparable (Wu et al., 2023, Lemma 3.1). Define

$$\mathcal{H}(\bar{\mathbf{w}}) := \frac{1}{n} \sum_{i \in \mathcal{S}} \exp(-y_i \bar{\mathbf{x}}_i^\top \bar{\mathbf{w}}),$$

then $\mathcal{H}(\cdot)$ is convex, bounded from below, and with a compact level set. Thus, it admits a finite minimizer, which is denoted as $\bar{\mathbf{w}}_* := \arg\min \mathcal{H}(\cdot)$.

Wu et al. (2025a, Lemma D.2) provided an asymptotic characterization of $\mathbf{w}_\lambda$, which is restated as the following lemma.

**Lemma 19** (Lemma D.2 in (Wu et al., 2025a)). *Under Assumption 3, as $\lambda \to 0$, we have*

$$w_\lambda \to \infty, \quad \bar{\mathbf{w}}_\lambda \to \bar{\mathbf{w}}_*.$$

Without loss of generality, we assume that $\{\mathbf{x}_1, \ldots, \mathbf{x}_n\}$ spans the whole space $\mathbb{H}$. Otherwise, we project every quantity into the span of $\{\mathbf{x}_1, \ldots, \mathbf{x}_n\}$. Under this convention, we have the following sharp characterization of the Hessian at $\mathbf{w}_\lambda$.

**Lemma 20** (Hessian bounds). *Under Assumption 3, as $\lambda \to 0$, we have*

$$\nabla^2 \mathcal{L}(\mathbf{w}_\lambda) = \lambda \ln(1/\lambda) \frac{1 \pm o(1)}{\gamma^2 \mathcal{H}(\bar{\mathbf{w}}_*)} \frac{1}{n} \sum_{i \in \mathcal{S}} \mathbf{x}_i \mathbf{x}_i^\top \exp(-y_i \bar{\mathbf{x}}_i^\top \bar{\mathbf{w}}_*).$$

*As a direct consequence, for every $\lambda < 1/C_0$, we have*

$$\frac{1}{C_1} \lambda \ln(1/\lambda) \mathbf{I} \preceq \nabla^2 \mathcal{L}(\mathbf{w}_\lambda) \preceq C_1 \lambda \ln(1/\lambda) \mathbf{I},$$

*where $C_0, C_1 > 1$ depend on the dataset but are independent of $\lambda$.*

*Proof of Lemma 20.* The first-order optimality condition for $\mathbf{w}_\lambda$ implies that

$$\lambda w_\lambda = \frac{1}{n} \sum_{i=1}^{n} \frac{y_i x_i}{1 + \exp(y_i \mathbf{x}_i^\top \mathbf{w}_\lambda)}$$

$$= \frac{1}{n} \sum_{i \in \mathcal{S}} \frac{\gamma}{1 + \exp(\gamma w_\lambda + y_i \bar{\mathbf{x}}_i^\top \bar{\mathbf{w}}_\lambda)} + \frac{1}{n} \sum_{i \notin \mathcal{S}} \frac{y_i x_i}{1 + \exp(y_i x_i w_\lambda + y_i \bar{\mathbf{x}}_i^\top \bar{\mathbf{w}}_\lambda)},$$

where $y_i x_i > \gamma$ for $i \notin \mathcal{S}$. Then Lemma 19 implies that

$$\lambda w_\lambda \exp(\gamma w_\lambda) = \frac{1}{n} \sum_{i \in \mathcal{S}} \frac{\gamma \exp(\gamma w_\lambda)}{1 + \exp(\gamma w_\lambda + y_i \bar{\mathbf{x}}_i^\top \bar{\mathbf{w}}_\lambda)} + \frac{1}{n} \sum_{i \notin \mathcal{S}} \frac{y_i x_i \exp(\gamma w_\lambda)}{1 + \exp(y_i x_i w_\lambda + y_i \bar{\mathbf{x}}_i^\top \bar{\mathbf{w}}_\lambda)}$$

$$= \left(1 + o(1)\right) \frac{1}{n} \sum_{i \in \mathcal{S}} \frac{\gamma \exp(\gamma w_\lambda)}{1 + \exp(\gamma w_\lambda + y_i \bar{\mathbf{x}}_i^\top \bar{\mathbf{w}}_\lambda)}$$

$$= \left(1 \pm o(1)\right) \frac{1}{n} \sum_{i \in \mathcal{S}} \frac{\gamma \exp(\gamma w_\lambda)}{\exp(\gamma w_\lambda + y_i \bar{\mathbf{x}}_i^\top \bar{\mathbf{w}}_\lambda)}$$

$$= \left(1 \pm o(1)\right) \gamma \mathcal{H}(\bar{\mathbf{w}}_\lambda)$$

$$= \left(1 \pm o(1)\right) \gamma \mathcal{H}(\bar{\mathbf{w}}_*).$$

That is,

$$\gamma w_\lambda \exp(\gamma w_\lambda) = \left(1 \pm o(1)\right) \gamma^2 \mathcal{H}(\bar{w}_*)/\lambda.$$

Notice that $\gamma w_\lambda$ is the Lambert W function applied to the right-hand side. By the property of the Lambert W function(see, e.g., Hoorfar and Hassani, 2008, Theorem 2.7), we have

$$\exp(\gamma w_\lambda) = \left(1 \pm o(1)\right) \frac{(1 \pm o(1)) \gamma^2 \mathcal{H}(\bar{\mathbf{w}}_*)/\lambda}{\ln\left((1 \pm o(1)) \gamma^2 \mathcal{H}(\bar{\mathbf{w}}_*)/\lambda\right)} = \left(1 \pm o(1)\right) \frac{\gamma^2 \mathcal{H}(\bar{\mathbf{w}}_*)}{\lambda \ln(1/\lambda)}.$$

For the Hessian at $\mathbf{w}_\lambda$, we have

$$\nabla^2 \mathcal{L}(\mathbf{w}_\lambda) = \frac{1}{n} \sum_{i=1}^{n} \frac{\mathbf{x}_i \mathbf{x}_i^\top}{(1 + \exp(-y_i \mathbf{x}_i^\top \mathbf{w}_\lambda))(1 + \exp(y_i \mathbf{x}_i^\top \mathbf{w}_\lambda))}$$

$$= \left(1 \pm o(1)\right) \frac{1}{n} \sum_{i=1}^{n} \frac{\mathbf{x}_i \mathbf{x}_i^\top}{\exp(y_i x_i w_\lambda + y_i \bar{\mathbf{x}}_i^\top \bar{\mathbf{w}}_\lambda)}$$

$$= \left(1 \pm o(1)\right) \frac{1}{n} \sum_{i \in \mathcal{S}} \frac{\mathbf{x}_i \mathbf{x}_i^\top}{\exp(\gamma w_\lambda + y_i \bar{\mathbf{x}}_i^\top \bar{\mathbf{w}}_\lambda)}$$

$$= \left(1 \pm o(1)\right) \exp(-\gamma w_\lambda) \frac{1}{n} \sum_{i \in \mathcal{S}} \exp(-y_i \bar{\mathbf{x}}_i^\top \bar{\mathbf{w}}_\lambda) \mathbf{x}_i \mathbf{x}_i^\top$$

$$= \left(1 \pm o(1)\right) \exp(-\gamma w_\lambda) \frac{1}{n} \sum_{i \in \mathcal{S}} \exp(-y_i \bar{\mathbf{x}}_i^\top \bar{\mathbf{w}}_*) \mathbf{x}_i \mathbf{x}_i^\top.$$

Plugging in the bounds for $\exp(\gamma w_\lambda)$, we get

$$\nabla^2 \mathcal{L}(\mathbf{w}_\lambda) = \lambda \ln(1/\lambda) \frac{1 \pm o(1)}{\gamma^2 \mathcal{H}(\bar{\mathbf{w}}_*)} \frac{1}{n} \sum_{i \in \mathcal{S}} \mathbf{x}_i \mathbf{x}_i^\top \exp(-y_i \bar{\mathbf{x}}_i^\top \bar{\mathbf{w}}_*),$$

which concludes the proof. □

We are ready to prove Theorem 7.

*Proof of Theorem 7.* Lemma 20 implies that for every $\lambda \leq 1/C_0$, we have

$$\frac{1}{C_1} \lambda \ln(1/\lambda) \mathbf{I} \preceq \nabla^2 \widetilde{\mathcal{L}}(\mathbf{w}_\lambda) := \lambda \mathbf{I} + \nabla^2 \mathcal{L}(\mathbf{w}_\lambda) \preceq C_1 \lambda \ln(1/\lambda) \mathbf{I}.$$

Since $\nabla^2 \widetilde{\mathcal{L}}(\cdot)$ is continuously differentiable, there exists a neighborhood of $\mathbf{w}_\lambda$ of radius $r$ such that

for all $\mathbf{w}$ such that $\|\mathbf{w} - \mathbf{w}_\lambda\| \leq r$, $\quad \frac{1}{2C_1} \lambda \ln(1/\lambda) \mathbf{I} \preceq \nabla^2 \widetilde{\mathcal{L}}(\mathbf{w}) \preceq 2C_1 \lambda \ln(1/\lambda) \mathbf{I}.$

**The first claim.** By classical optimization theory, GD with initialization satisfying $\|\mathbf{w}_0 - \mathbf{w}_\lambda\| \le r$ and stepsize satisfying $\eta < 1/(C_1 \lambda \ln(1/\lambda))$ converges to $\mathbf{w}_\lambda$.

**The second claim.** Recall that $\nabla\widetilde{\mathcal{L}}(\mathbf{w}_\lambda) = 0$. For every $\mathbf{w}$ such that $\|\mathbf{w} - \mathbf{w}_\lambda\| \le r$,

$$\nabla\widetilde{\mathcal{L}}(\mathbf{w}) = \int_0^1 \nabla^2\widetilde{\mathcal{L}}(t\mathbf{w} + (1-t)\mathbf{w}_\lambda)(\mathbf{w} - \mathbf{w}_\lambda)dt.$$

This, together with the above Hessian bound, implies that

$$\frac{1}{2C_1}\lambda\ln(1/\lambda)\|\mathbf{w} - \mathbf{w}_\lambda\| \le \|\nabla\widetilde{\mathcal{L}}(\mathbf{w})\| \le 2C_1\lambda\ln(1/\lambda)\|\mathbf{w} - \mathbf{w}_\lambda\|.$$

Consider GD with a stepsize $\eta > 20C_1^3/(\lambda\ln(1/\lambda))$. If GD is within that ball in the $t$-th step, then we have

$$\|\mathbf{w}_{t+1} - \mathbf{w}_\lambda\|^2 = \|\mathbf{w}_t - \mathbf{w}_\lambda\|^2 - 2\eta\langle\nabla\widetilde{\mathcal{L}}(\mathbf{w}_t), \mathbf{w}_t - \mathbf{w}_\lambda\rangle + \eta^2\|\nabla\widetilde{\mathcal{L}}(\mathbf{w}_t)\|^2$$

$$\ge \|\mathbf{w}_t - \mathbf{w}_\lambda\|^2 - 4\eta C_1\lambda\ln(1/\lambda)\|\mathbf{w}_t - \mathbf{w}_\lambda\|^2 + \left(\frac{1}{2C_1}\eta\lambda\ln(1/\lambda)\right)^2\|\mathbf{w}_t - \mathbf{w}_\lambda\|^2$$

$$\ge \|\mathbf{w}_t - \mathbf{w}_\lambda\|^2 + \eta C_1\lambda\ln(1/\lambda)\|\mathbf{w}_t - \mathbf{w}_\lambda\|^2$$

$$\ge (1 + 20C_1^4)\|\mathbf{w}_t - \mathbf{w}_\lambda\|^2.$$

That is, if GD enters the ball centered at $\mathbf{w}_\lambda$ with radius $r$ but is different from $\mathbf{w}_\lambda$, then it must exit the ball in a finite number of steps. However, we will show next that the set of the initializations such that GD exactly hits $\mathbf{w}_\lambda$ has measure zero.

Let $d < \infty$ be the dimension of $\mathbb{H}$, then we can embed $\mathbb{H}$ into $\mathbb{R}^d$. Let

$$g : \mathbb{R}^d \to \mathbb{R}^d, \quad \mathbf{w} \mapsto \mathbf{w} - \eta\nabla\widetilde{\mathcal{L}}(\mathbf{w})$$

be one step of GD. To conclude, it suffices to show that $g$ satisfies the Luzin $N^{-1}$ property, that is, for all subsets $S \subset \mathbb{R}^d$, if $S$ has measure zero then its preimage $g^{-1}(S)$ also has measure zero. Indeed, this ensures that (countably infinite times) iterated preimages of $\{\mathbf{w}_\lambda\}$ remain of measure zero. Conveniently, showing $g$ satisfies the Luzin $N^{-1}$ property is equivalent to showing that the Jacobian determinant of $g$ is nonzero almost everywhere (Ponomarev, 1987, Theorem 1). We denote the Jacobian determinant of $g$ as

$$\Delta : \mathbb{R}^d \to \mathbb{R}, \quad \mathbf{w} \mapsto \det\left(\mathbf{I} - \eta\nabla^2\widetilde{\mathcal{L}}(\mathbf{w})\right).$$

Observe that $\Delta$ is a composition of a degree-$d$ polynomial, of the derivatives of the sigmoid function $x \mapsto 1/(1+e^{-x})$, and of linear maps of $\mathbf{w}$. Recall that the sigmoid function is analytic on $\mathbb{R}$, meaning that it is everywhere equal to its Taylor expansion on a ball of positive radius. We conclude that $\Delta$ is also analytic on $\mathbb{R}^d$ as a composition of analytic functions. By the identity theorem for analytic functions (Krantz and Parks, 2002, Corollary 1.2.7), we conclude that $\Delta$ is either zero everywhere or that its zeros do not have an accumulation point in $\mathbb{R}^d$. We show that the latter holds by discussing the following two cases.

- If $\eta = 1/\lambda$, then

$$\Delta(\mathbf{0}) = \det\left((1 - \eta\lambda)\mathbf{I} - \eta\frac{1}{4n}\sum_{i=1}^n \mathbf{x}_i\mathbf{x}_i^\top\right) = \det\left(-\eta\frac{1}{4n}\sum_{i=1}^n \mathbf{x}_i\mathbf{x}_i^\top\right),$$

  which is nonzero since we assume $\{\mathbf{x}_1, \ldots, \mathbf{x}_n\}$ spans the whole space.

- If $\eta \ne 1/\lambda$, then by Assumption 1, as $\rho \to \infty$, we have

$$\Delta(\rho\mathbf{w}^*) = \det\left((1 - \eta\lambda)\mathbf{I} - \eta\frac{1}{n}\sum_{i=1}^n \frac{\mathbf{x}_i\mathbf{x}_i^\top}{\left(1 + \exp(\rho\mathbf{x}_i^\top\mathbf{w}^*)\right)\left(1 + \exp(-\rho\mathbf{x}_i^\top\mathbf{w}^*)\right)}\right) \to 1 - \eta\lambda.$$

  That is, for every pair of $\eta$ and $\lambda$ such that $\eta\lambda \ne 1$, we can pick a sufficiently large $\rho$ such that $\Delta(\rho\mathbf{w}^*) = 1 - \eta\lambda \pm o(1)$ is nonzero.

In sum, for any choices of $\eta$ and $\lambda$, $\Delta$ cannot be zero everywhere. So the zeros of $\Delta$ do not have an accumulation point in $\mathbb{R}^d$, and thus have measure zero. This concludes the proof. $\qquad\square$

## D.2 Global convergence in the 1-dimensional case

In the 1-dimensional case, the objective function can be written as

$$\widetilde{\mathcal{L}}(w) := \frac{1}{n} \sum_{i=1}^{n} \ln(1 + e^{-z_i w}) + \frac{\lambda}{2} w^2 \, ,$$

where $\gamma \leq z_i \leq 1$ and there exists an $i$ such that $z_i = \gamma$.

The next theorem shows that in this 1-dimensional case, GD converges globally with stepsizes below the critical threshold by a constant factor. We note that this is a very special situation, where GD with large stepsizes oscillates *at most once* (see the proof). However, in general finite-dimensional cases, GD with large stepsizes can oscillate many times. Thus, it is unclear if the results in this theorem generalize to general finite-dimensional cases.

**Theorem 8** (A 1-dimensional anlaysis). *Suppose that Assumption 1 holds and that $\mathbb{H}$ is 1-dimensional. Then for every $\lambda \leq 1/C_0$, $\eta \leq 1/(C_1 \lambda \ln(1/\lambda))$, and $w_0$, GD converges, and after*

$$t = \mathcal{O}\left( \frac{1}{\eta \lambda} \ln \left( \frac{|w_0| + 1}{\varepsilon \lambda \ln(1/\lambda)} \right) \right)$$

*steps, $\widetilde{\mathcal{L}}(w_t) - \min \widetilde{\mathcal{L}} \leq \varepsilon$. Here, $C_0, C_1 > 1$ depend on the dataset and on $\gamma$, but not on $\lambda$ or $\eta$.*

*Proof of Theorem 8.* Let us first compute the derivatives of the objective,

$$\widetilde{\mathcal{L}}'(w) = -\frac{1}{n} \sum_{i=1}^{n} \frac{z_i}{1 + e^{z_i w}} + \lambda w, \quad \widetilde{\mathcal{L}}''(w) = \frac{1}{n} \sum_{i=1}^{n} \frac{z_i^2}{(1 + e^{-z_i w})(1 + e^{z_i w})} + \lambda.$$

Setting $\widetilde{\mathcal{L}}'(w_\lambda) = 0$ and using the same argument as the proof of Lemma 20 shows that

$$\exp(\gamma w_\lambda) = (1 \pm o(1)) \frac{\gamma^2 p}{\lambda \ln(1/\lambda)},$$

where $p = |\{i : z_i = \gamma\}|/n$. Then for any $w \geq w_\lambda - 1$, we have

$$\widetilde{\mathcal{L}}''(w) = \frac{1}{n} \sum_{i=1}^{n} \frac{z_i^2}{(1 + e^{-z_i w})(1 + e^{z_i w})} + \lambda$$

$$\leq \frac{1}{n} \sum_{i=1}^{n} \frac{1}{(1 + e^{z_i w})} + \lambda$$

$$\leq \frac{1}{n} \sum_{i=1}^{n} e^{-z_i w} + \lambda$$

$$\leq e^{\gamma} \exp(-\gamma w_\lambda) + \lambda$$

$$= (1 \pm o(1)) \frac{e}{\gamma^2 p} \lambda \ln(1/\lambda) + \lambda$$

$$\leq C \lambda \ln(1/\lambda),$$

for sufficiently small $\lambda$ and $C$ that depends on $p$ and $\gamma$. Moreover, observe that $\widetilde{\mathcal{L}}''(w) > \lambda$ for all $w$.

Observe that $\widetilde{\mathcal{L}}'(\cdot)$ is increasing and that $\widetilde{\mathcal{L}}'(w_\lambda) = 0$, so we have

$$\widetilde{\mathcal{L}}'(w) \begin{cases} \leq 0 & w \leq w_\lambda \\ \geq 0 & w \geq w_\lambda. \end{cases}$$

For any $w$, by Taylor's theorem, since $\widetilde{\mathcal{L}}'(w_\lambda) = 0$, there exists a $v$ between $w$ and $w_\lambda$ such that

$$\widetilde{\mathcal{L}}'(w) = \widetilde{\mathcal{L}}''(v)(w - w_\lambda).$$

If $w \geq w_\lambda$, $v \geq w_\lambda$ and so this implies

$$\lambda(w - w_\lambda) \leq \widetilde{\mathcal{L}}'(w) \leq C \lambda \ln(1/\lambda)(w - w_\lambda).$$

Alternatively, if $w_\lambda - 1 \leq w \leq w_\lambda$, it implies $\widetilde{\mathcal{L}}'(w) \geq C \lambda \ln(1/\lambda)(w - w_\lambda)$. Finally, for any $w \leq w_\lambda$, it implies

$$-1 < \widetilde{\mathcal{L}}'(w) \leq \lambda(w - w_\lambda).$$

Let $t_0$ be the first time such that $w_t > w_\lambda$; note that $t_0$ might be infinite.

**Consider $t < t_0$.** By definition, we have $w_t \leq w_\lambda$ for all $t < t_0$. Thus

$$0 \geq w_t - w_\lambda = w_{t-1} - w_\lambda - \eta \widetilde{\mathcal{L}}'(w_{t-1})$$
$$\geq w_{t-1} - w_\lambda - \eta\lambda(w_{t-1} - w_\lambda) = (1 - \eta\lambda)(w_{t-1} - w_\lambda).$$

This implies that $|w_t - w_\lambda| \leq (1 - \eta\lambda)^t |w_0 - w_\lambda|$ for $t < t_0$. Hence, for

$$t \geq \frac{1}{\eta\lambda} \ln(|w_0| + w_\lambda),$$

either $t \geq t_0$ or $w_\lambda - 1 \leq w_{t-1} \leq w_\lambda$. In the latter case

$$w_t - w_\lambda = w_{t-1} - w_\lambda - \eta\widetilde{\mathcal{L}}(w_{t-1}) \leq (w_{t-1} - w_\lambda)\left(1 - C\eta\lambda\ln(1/\lambda)\right) \leq 0,$$

provided $C_1$ is chosen sufficiently large. And in this case, $w_t \leq w_\lambda$ for all subsequent $t$. That is, either $t_0$ is infinite, in which case the step complexity is $O(\ln((|w_0| + w_\lambda)/\varepsilon)/(\eta\lambda))$, or

$$t_0 \leq \frac{1}{\eta\lambda} \ln(|w_0| + w_\lambda).$$

**Consider $t \geq t_0$.** By definition we have $w_{t_0} > w_\lambda$. We show $w_t \geq w_\lambda$ for all $t \geq t_0$ by induction. Recall the stepsize condition that $\eta < 1/(2C\lambda\ln(1/\lambda))$. Assume that $w_t \geq w_\lambda$, then

$$w_{t+1} = w_t - \eta\widetilde{\mathcal{L}}'(w_t) \geq w_t - \eta C\lambda\ln(1/\lambda)(w_t - w_\lambda) \geq w_t - \frac{1}{2}(w_t - w_\lambda) \geq \frac{1}{2}(w_t + w_\lambda) \geq w_\lambda.$$

So by induction, we have $w_t \geq w_\lambda$ for all $t \geq t_0$. Also,

$$0 \leq w_{t+1} - w_\lambda = w_t - w_\lambda - \eta\widetilde{\mathcal{L}}'(w_t) \leq w_t - w_\lambda - \eta\lambda(w_t - w_\lambda) = (1 - \eta\lambda)(w_t - w_\lambda).$$

That is, $|w_t - w_\lambda| \leq (1 - \eta\lambda)^{t-t_0} |w_{t_0} - w_\lambda|$ for $t \geq t_0$. Finally, notice that

$$w_\lambda < w_{t_0} \leq w_{t_0 - 1} + \eta \leq w_\lambda + \eta,$$

so $|w_{t_0} - w_\lambda| \leq \eta = \Theta(1/(\lambda\ln(1/\lambda)))$. So we have $|w_t - w_\lambda| \leq (1 - \eta\lambda)^{t-t_0}\Theta(1/(\lambda\ln(1/\lambda)))$ for $t \geq t_0$.

Combining with the bound on $t_0$ shows that the step complexity is

$$O\left(\frac{1}{\eta\lambda} \ln\left(\frac{|w_0| + \ln(1/(\lambda\ln(1/\lambda)))}{\varepsilon\lambda\ln(1/\lambda)}\right)\right) = O\left(\frac{1}{\eta\lambda} \ln\left(\frac{|w_0| + 1}{\varepsilon\lambda\ln(1/\lambda)}\right)\right).$$

This completes our proof. $\qquad\square$

# E  Experimental details

The dataset is composed on two datapoints $x_1 = (\gamma, 1)$ and $x_2 = (\gamma, -2)$ for $\gamma = 0.2$. We run GD on the regularized logistic regression for $\lambda = 2^{-12}$, a logarithmic range of stepsizes from $2^1$ to $2^{13}$, and $2^{13}$ steps. Additional plots are given in Figure 3. Two comments are of interest. First, we observe that the dynamics converge for stepsizes up to $2^5$. This is consistent with the local stability threshold given by $2/\|\nabla\widetilde{L}(\mathbf{w}_\lambda)\|_2 \approx 44.8$. Second, we observe in the case of $\eta = 2^5$ that even after $\widetilde{L}$ stops oscillating for $t \approx 2^4$ (start of the stable phase), both the regularization and the logistic components continue to evolve nonmonotonically. This is connected to the discussion in Section 2.3, where we outline that a decrease of $\widetilde{\mathcal{L}}(\mathbf{w})$ may cause an increase of $\mathcal{L}(\mathbf{w})$, and then GD might leave the stable region.

The code was implemented in JAX (Bradbury et al., 2018) and takes a few seconds to run on a consumer laptop. Our code is available at https://github.com/PierreMarion23/large-stepsize-regularized-logistic.

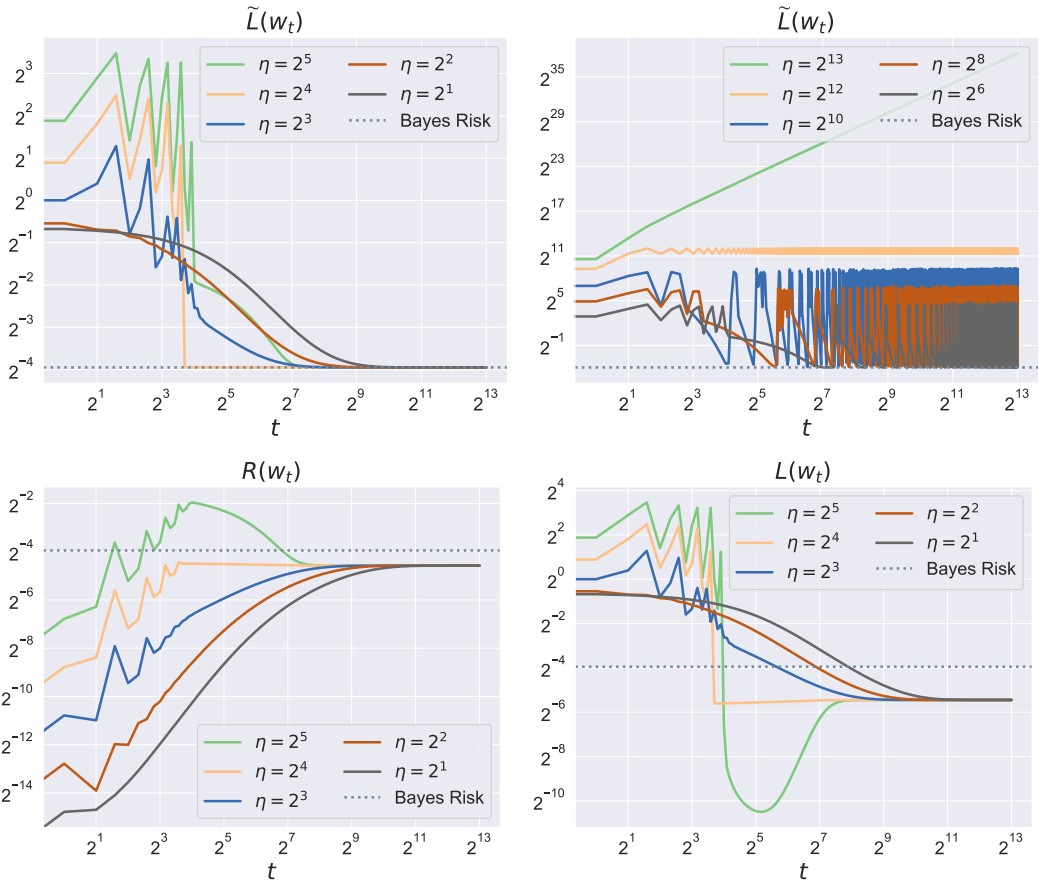

Figure 3: Additional plots for the 2-dimensional experiment. **Top left:** Objective value as a function of training steps. **Top right:** Objective value as a function of training steps for even larger stepsizes. **Bottom left:** Value of the regularization component as a function of training steps. **Bottom right:** Value of the logistic component as a function of training steps.

