# OpenReview forum: "Large Stepsizes Accelerate Gradient Descent for Regularized Logistic Regression"
_NeurIPS.cc/2025/Conference — NeurIPS 2025 poster_

### Official Review · Reviewer_Lvra · 2025-06-30

**Clarity:** 4
**Significance:** 3
**Originality:** 3
**Rating:** 5
**Confidence:** 3

**Summary:**

This paper studies gradient descent (GD) for $\ell_2$-regularized logistic regression with linearly separable data under a large constant stepsize. Considering different regimes of the regularization strength, this paper shows various accelerating behavior induced by the large stepsize, particularly about the jump between stable phase (where the loss decreases monotonically) and edge of stability (EoS) phase (the loss does not decrease monotonically) where phase transition points are indentified. Moreover, the critical threshold on the convergent stepsizes is also discussed.

**Questions:**

1. Can you discuss more on the assumptions? Does similar result hold if the separability condition is not satisfied?

2. What is the explicit dependence of the constants $C_1,C_2,C_3$ in the theorems 1, 4 and 7 on the dataset? How large will they be?

3. Can the result and techniques be extended to other models such as generalized linear regression?

**Ethical Concerns:**

["NO or VERY MINOR ethics concerns only"]

**Final Justification:**

This paper is technically solid and provides novel and interesting theoretical results.

**Limitations:**

yes

**Quality:**

3

**Strengths And Weaknesses:**

### Strengths
* This paper is well written and clear to read.
* This paper studies the interplay of large stepsize and EoS for $\ell_2$-regularized logistic regression, which is novel.
The result on the different regimes of stepsizes is very interesting and provide more understandings for the study of EoS.
* The theorems in this paper are clear and seem to be well supported by the proofs (though I haven't checked all the details).


### Weaknesses

* Numerical experiments can be included to further justify the theoretical results. Demonstrating a training path can provide readers with more intuitions.
* Filling out the missing parts of the full picture of the theoretical results as shown in Figure 1 would make the theory more complete.

---

> ### Author Rebuttal · Authors · 2025-07-29
>
> Thank you for supporting our paper! We address your comments as follows.
>
> ---
> **Q1.** “Numerical experiments can be included to further justify the theoretical results. Demonstrating a training path can provide readers with more intuitions.”
>
> **A1.** As our results are of theoretical nature, we don’t think experiments are necessary for justifying our results. With that being said, we agree with the reviewer that the suggested simulations are helpful for providing intuitions. We will add simulations, including visualizations of a training path, in the revision.
>
> ---
> **Q2.** “Filling out the missing parts of the full picture of the theoretical results as shown in Figure 1 would make the theory more complete.”
>
> **A2.** We agree that there are open problems to be solved as commented throughout the paper. Yet, our current results, which show the benefits of large stepsizes for GD for regularized logistic regression, are already significant.
>
> ---
> **Q3.** “Can you discuss more on the assumptions? Does similar result hold if the separability condition is not satisfied?”
>
> **A3.** As mentioned in Lines 99-109, linear separability is a standard assumption in the theoretical literature, so it is not restrictive.
>
> Moreover, our results can be extended to certain cases with non-linearly-separable data. For example, our results can be extended to wide neural networks in the lazy regime for any finite number data in the general position (see the discussion in Lines 361-362). Note that any finite number data in the general position is guaranteed to be separable in the RKHS given by the neural tangent kernel, even if they are not linearly separable.
>
> For linear predictors with general non-linearly-separable dataset, it is unclear if our results hold. We will comment on this as a future work in the revision.
>
> ---
> **Q4.** “What is the explicit dependence of the constants $C_1, C_2, C_3$  in the theorems 1, 4 and 7 on the dataset? How large will they be?”
>
> **A4.** In Theorems 1 and 4, the constants are all numerical and do not depend on anything including the dataset. In Theorem 7, the quantities $C_1$ and $C_2$ depend on the dataset through the margin $\gamma$ and the second derivative of the loss at the minimizer (detailed in Lemma 20 in Appendix D.1).
>
> ---
> **Q5.** “Can the result and techniques be extended to other models such as generalized linear regression?”
>
> **A5.** Part of our results and analysis can be extended to certain generalized linear models. For example, for linear predictors with general loss functions given by Assumption 3 in (Wu et al. 2024). We have commented on this and some other possible extensions in Lines 360-363.

---

> > ### Comment · Reviewer_Lvra · 2025-08-01
> >
> > Thank you for the response, I will keep my score.

---

### Official Review · Reviewer_ieJz · 2025-07-03

**Clarity:** 3
**Significance:** 3
**Originality:** 3
**Rating:** 5
**Confidence:** 4

**Summary:**

This paper shows that gradient descent (GD) with a properly chosen large constant step size achieves fast convergence for $\ell_2$-regularized logistic regression when the data is linearly separable. This thereby extends our understanding of the optimization benefit of GD operating at the edge of stability regime. In particular, this work builds on a recent result by Wu et al. (2024), which showed that a large constant step size can accelerate GD in the unregularized logistic regression setting. Although the two results appear related, the proof techniques developed in the unregularized setting do not directly extend to the regularized case considered here, due to the presence of a finite minimizer.

For a small regularization parameter $\lambda = O(1/n^2)$, Theorem 1 establishes that large step GD achieves a convergence rate of $O(\ln(1/\epsilon)/\sqrt{\lambda})$ matching that of Nesterov's momentum. Moreover, the analysis shows that GD enters a stable phase with linear convergence after a finite number of steps. While the acceleration is theoretically appealing, the authors correctly note that such a small regularization level may be inadequate in the presence of statistical noise.

To address this limitation, this paper further investigates the regime of larger regularization and statistical uncertainty. When $\lambda=O(1)$ and the step size is set to $\eta=O(1/\lambda^{1/3})$, GD achieves the complexity of $O\(\ln(1/\epsilon)/\lambda^{2/3})$, which improves over small-step GD but remains slower than Nesterov's momentum. In Section 3, the authors further investigate the benefits of large step under statistical uncertainty. Based on the best-known population risk upper bound in Proposition 6, the authors argue that the largest reduction in population risk is obtained when choosing $\lambda=1/n$ and $\eta=(\gamma^2n)^{1/3}$. However, under this setup, the rate does not match the fast rate provided by Nesterov's momentum.

Finally, this paper completes its analysis by establishing that $1/\sqrt{\lambda}$ is the critical threshold between (local) convergence and divergence of large-step GD, under a mild support vectors condition.

**Questions:**

- Page 2 line 64: Is there a reference that supports this statement? If this claim is supported by this paper, I think it would be helpful to add a brief explanation or clarification at this point.
- Page 3 line 117: Why is there a gap in the large step size analysis between the separable and non-separable cases? Is it due to logistic regression being strictly convex but not strongly convex under strictly non-separable data? What would happen if one adds regularization in the strictly non-separable case?
- Page 4 line 151: Could you provide some intuition behind the transition in behavior at the step size $1/\sqrt{\lambda}$?
- Page 8 Table 1: Do the theoretical comparisons in Table 1 align with the empirical observations?
- Page 9 line 365: Could you clarify the last two future directions? They were somewhat vague and difficult to interpret.

- Minor comments: Unlike the main texts, the conclusion does not explicitly mention the requirement of small regularization for matching the rate of Nesterov's momentum. I suggest including this. In addition, when referring to (Cai et al., 2024), something other than linearly separable data should be mentioned.

**Ethical Concerns:**

["NO or VERY MINOR ethics concerns only"]

**Limitations:**

The paper clearly states its limitations at the end of each section.

**Quality:**

3

**Strengths And Weaknesses:**

- S1: As noted in Section 2.3, while the main claims resemble those of Wu et al. (2024), the proof technique developed in this paper for the regularized setting is novel and nontrivial.
- S2: Despite the limited benefit of large step size under strong regularization and statistical uncertainty, this paper offers a comprehensive theoretical analysis of large step GD for regularized logistic regression.
- S3: This paper is clearly written and well-structured. The main results appear technically sound, although I have not carefully checked the proofs.

- W1: Related to S2, the rate matching the optimal rate of Nesterov's momentum is achieved only in the small regularization regime, which limits the practical relevance of the result under statistical uncertainty.

---

> ### Author Rebuttal · Authors · 2025-07-29
>
> Thanks for supporting our submission! Please find our answers to your comments below.
>
> ---
> **Q1.** “Related to S2, the rate matching the optimal rate of Nesterov's momentum is achieved only in the small regularization regime, which limits the practical relevance of the result under statistical uncertainty.”
>
> **A1.** We emphasize that for a large regularization, we have also shown an improvement of using large stepsizes over small ones for GD. Although the improvement does not match Nesterov’s acceleration, we don’t feel this limits the practical relevance of our theory. In fact, it is unclear to us if large stepsizes GD can also match Nesterov’s acceleration for a large regularization. We will comment on this as a concrete open problem in the revision.
>
> ---
> **Q2.** “Page 2 line 64: Is there a reference that supports this statement? If this claim is supported by this paper, I think it would be helpful to add a brief explanation or clarification at this point.”
>
> **A2.** This is an informal statement for providing intuition for categorizing convergent GD runs into the stable regime and the EoS regime. This statement holds under weak regularity assumptions, e.g., when the objective is second-order continuously differentiable and strongly convex, and the stepsize satisfies a minimum condition for local stability, $\\eta < 2/ \\\| \\nabla\^2 L(\\hat w) \\\|$, where $\\hat w$ is the global minimizer. In this case, there exists a neighbor of $\\hat w$ such that GD is stable within this neighborhood. As any convergent GD run has to enter this neighbor, it has to enter a stable phase in finite steps. We will clarify this in the revision.
>
> ---
> **Q3.** “Page 3 line 117: Why is there a gap in the large step size analysis between the separable and non-separable cases? Is it due to logistic regression being strictly convex but not strongly convex under strictly non-separable data? What would happen if one adds regularization in the strictly non-separable case?”
>
> **A3.** Firstly, we’d like to clarify that for both separable data and strictly non-separable data, logistic regression is strictly convex but non-strongly-convex, as the domain is unbounded.
>
> Secondly, our results are not directly comparable to those of Meng et al. (2024). Indeed, their counter example is for stepsizes in the order of $1/\\\|\\nabla\^2 L(\\hat w)\\\|$, where $\hat w$ is the global minimizer. This corresponds to $\eta = \Theta(1 / (\lambda \ln(1/\lambda) ) )$ in our settings. In comparison, our theory requires $\eta = O(1/\sqrt{\lambda})$, which is much smaller.
>
> Finally, it is unclear to us if GD always converges with $\eta = \Theta(1 / (\lambda \ln(1/\lambda) ) )$ even under separable data. We have commented on this as Conjecture 1 in Lines 350-352.
>
> ---
> **Q4.** “Page 4 line 151: Could you provide some intuition behind the transition in behavior at the step size 1/sqrt(lambda) ?”
>
> **A4.** The stepsize $\eta = \Theta(1/\sqrt{\lambda})$ is the largest one for which our proof is valid. However, preliminary experiments suggest that convergence may still occur for larger stepsizes, and characterizing this behavior is an interesting open problem (as we emphasize at the end of Section 4). Technically, our proof fails at Lemma 13. While pinpointing the exact place of failure requires getting into somewhat technical computations, the critical scaling $\eta = \Theta(1/\sqrt{\lambda})$ can be seen by inspecting Line 895: the squared gradient norm inside the parenthesis is less than 1, while $\\\|w\_t\\\|\^2$ can be of order $\eta^2$. Then, if $\eta$ exceeds the critical scaling, the expression in the parentheses becomes strictly negative, rendering the inequality uninformative.
>
> ---
> **Q5.** “Page 8 Table 1: Do the theoretical comparisons in Table 1 align with the empirical observations?”
>
> **A5.** As discussed in Lines 315-319, the comparison in Table 1 relies on the best-known margin-based (worst-case) population risk bound in our setting. While we believe this bound is fairly tight for moderately large $n$, this bound could be loose for exponentially large $n$ (in terms of dimension and/or margin), as the population distribution is realizable (i.e., Bayes risk is zero). Furthermore, the bound might be also loose for average-case problems encountered in practice. We acknowledge this limitation of Table 1. We will conduct experiments to empirically compare the generalization of each method, and include our findings in the revision.
>
> ---
> **Q6.** “Page 9 line 365: Could you clarify the last two future directions? They were somewhat vague and difficult to interpret.”
>
> **A6.** In the second to the last future direction, we ask if there exists a natural statistical learning setting where large stepsizes lead to smaller population risk than small stepsizes. Note that we only show large stepsizes attain the same population risk faster than small stepsizes, but we haven’t shown large stepsizes attain a smaller population risk.
>
> In the last future direction, we ask if there is a generic approach to prove the convergence of large stepsize GD, without relying on the descent lemma. Note that our analysis is specific to logistic regression with linearly separable data (while it can be relaxed in certain ways), and fails to hold for a more general class of optimization problems.
>
> ---
> **Q7.** “Minor comments: Unlike the main texts, the conclusion does not explicitly mention the requirement of small regularization for matching the rate of Nesterov's momentum. I suggest including this. In addition, when referring to (Cai et al., 2024), something other than linearly separable data should be mentioned.”
>
> **A7.** Thank you for your suggestions. We will revise the conclusion to explicitly mention that we only match Nesterov’s acceleration when the regularization is small, and that (Cai et al., 2024) required linearly separable data and two-layer networks with fixed outer weights and strictly increasing activations such as leaky-ReLU.

---

### Official Review · Reviewer_q9wR · 2025-07-03

**Clarity:** 3
**Significance:** 2
**Originality:** 3
**Rating:** 3
**Confidence:** 3

**Summary:**

In the paper (Cohen et al., 2020), a phenomenon ``edge of stability’’ (EoS) regime is
investigated under the training of neural networks (NN). The EoS indicates that the NN
training can still converge if using a relatively large learning rate.

Another paper (Wu et al., 2024) utilizes a non-regularized logistic regression to investigate the
EoS regime. The current paper expand the results to regularized logistic regression
problems.

**Questions:**

See "weaknesses"

**Ethical Concerns:**

["NO or VERY MINOR ethics concerns only"]

**Final Justification:**

A solid theoretical contribution though I have some concerns about the practical impact.

**Limitations:**

yes

**Quality:**

3

**Strengths And Weaknesses:**

#### Strengths

- This paper is written clearly.
- This paper utilizes a simpler problem (logistic regression) to explain the Eos regime issue.

#### Weaknesses

- A concern is that this paper may have only theoretical values. For logistic regression, gradient descent is not an effective method. People have shown that quasi Newton (e.g., LBFGS), truncated Newton (via conjugat gradient methods) or even coordiate descent methods are better choices in different scenarios. Moreover, the paper concerns about linearly separable data, a situation not common in practice.

- The proofs in the appendices skip many details.
  Thus, it is hard to verify the correctness of the proofs.
  For example,
  a) In line 818, it seems that the right hand side lacks the term $$ -4 \eta \nabla L(w_t)^T w_t. $$
  b) In line 828, the left hand side is equal to $$ \eta^2 [ \nabla L (w_t)^T w^* + \| \nabla L(w_t) \|^2 ], $$
     but why this term is less than zero?
  c) In line 864, why do we have the second inequality
     $$ \frac{e}{n} \sum_{i=1}^{n} \left( \frac{1}{1+exp(y_i x_{i}^{T} w_t )} \right)^{1-\alpha \lambda \eta} \leq e \left( \frac{1}{n} \sum_{i=1}^{n} \frac{1}{1+exp(y_i x_{i}^{T} w_t)} \right)^{1-\alpha \lambda \eta} ? $$

- The paper does not contain any experiment to verify their proofs.

---

> ### Author Rebuttal · Authors · 2025-07-29
>
> We appreciate your detailed comments. We address your concerns below.
>
> ---
> **Q1.** “A concern is that this paper may have only theoretical values. For logistic regression, gradient descent is not an effective method. People have shown that quasi Newton (e.g., LBFGS), truncated Newton (via conjugat gradient methods) or even coordiate descent methods are better choices in different scenarios. Moreover, the paper concerns about linearly separable data, a situation not common in practice.”
>
> **A1.** We respectfully disagree with the assertion that “this paper may have only theoretical values”. Note that large stepsizes are the typical choice for using GD in machine learning, especially deep learning, as discussed in Lines 72-80. Our paper, by rigorously characterizing the benefits of large stepsizes in a natural setting, provides important insights into understanding why large stepsizes work so well in practice.
>
> We don’t think linearly separable data is restrictive, as logistic regression with linearly separable data is a standard setting in machine learning literature as discussed in Lines 99-113. Additionally, our results can be extended to neural networks in the lazy regime (see Line 362), in which the linear separability assumption is relaxed to separability in RKHS given by the neural tangent kernel, which holds for any finite number of data in general position.
>
> Finally, we emphasize that our goal is to understand the benefits of large stepsizes (over smaller ones) for GD. Although we use regularized logistic regression as a testbed, studying the fastest method for this problem is orthogonal to the topic of this paper.
>
> ---
> **Q2.** “The proofs in the appendices skip many details. Thus, it is hard to verify the correctness of the proofs…”
>
> **A2.** We’re fully confident with the correctness of our proof. We address each of your concerns below. Should the reviewer have any further questions, we’re happy to provide clarifications.
>
> * a) There is no missing term in the display below Line 818. Note that we have $u := u_1 + u_2 + u_3$, which implies that
> $$ 2 \eta \langle \nabla L(w_t), u - w_t \rangle = 2 \eta \langle \nabla L(w_t), u_1 - w_t \rangle + 2 \eta \langle \nabla L(w_t), u_2 \rangle + 2 \eta \langle \nabla L(w_t), u_3  \rangle$$
> The term with $ u_2$ cancels the gradient squared term, while the terms with $ u_1$ and $u_3$ are left.
>
> * b) The inequality below Line 828 is due to the same argument as the display below Line 817. We will clarify this in the revision.
>
> * c) The second step in the display below Line 864 is by Jensen’s inequality and the concavity of $x \mapsto x^{1 - \alpha \lambda \eta}$ as mentioned in Line 863.
>
>
> ---
> **Q3.** “The paper does not contain any experiment to verify their proofs.”
>
> **A3.** Our paper is of theoretical nature. All necessary assumptions and proofs are included for verifying the correctness of our theorems. It does not seem necessary to further verify our theorems with experiments. With that said, we are happy to include simulations in the revision to better illustrate our theory.

---

> > ### Comment · Reviewer_q9wR · 2025-08-04
> >
> > Thank you very much for addressing my comments. Some technical issues about the proof have been clarified.

---

> > > ### Author Response · Authors · 2025-08-04
> > >
> > > Thanks for confirming some of your concerns about our proof has been clarified. Feel free to follow up with any questions so we can address them during the rebuttal!

---

### Official Review · Reviewer_9WKU · 2025-07-07

**Clarity:** 3
**Significance:** 3
**Originality:** 3
**Rating:** 4
**Confidence:** 3

**Summary:**

This manuscript analyzes gradient descent with large constant step sizes for $\ell_2$-regularized logistic regression on linearly separable data. Building on a series of “edge-of-stability” (EoS) works that studied the unregularized case, the authors show that the same unstable dynamics can accelerate convergence in the strongly convex setting induced by $\ell_2$ regularization. Their choice of stepsize and regularization scaling is motivated by minimizing the generalization bound presented in Proposition 6.

Specifically, the manuscript shows that the iteration complexity depends on the scale of the regularization parameter $\lambda$. When $\lambda \leq O(n^{-2})$, gradient descent achieves $O(1/\sqrt{\lambda})$ iteration complexity, matching the convergence guarantee of Nesterov’s accelerated gradient method. For general $\lambda$, they obtain an iteration complexity of $O(\lambda^{-2/3})$.

In addition, the authors establish a lower bound on the iteration complexity when the loss function is monotone, showing that this regime requires $\Omega(1/\lambda)$ iterations, which demonstrates a separation between non monotonic and monotonic dynamics of GD.

**Questions:**

**Some discussions:**

* Both learning rates in Theorem 1 and Theorem 4 require a priori knowledge of the margin parameter $\gamma$. I would expect that if this parameter is not known, the generalization bounds in Table 1 could become suboptimal. Could you discuss how realistic this assumption is in practice? Additionally, it would be helpful to compare the results to adaptive gradient methods (e.g., Zhang et al. \[2025]) in the scenario where $\gamma$ is unavailable.
* The convergence rate predicted by\ Theorem 1 is worse than that of Theorem 4 in the range $O(n^{-3/2}) \leq \lambda \leq O(n^{-1})$, suggesting there may be room for improving the analysis of Theorem 1. What causes this discrepancy? Clarifying this point could be helpful for readers.

**Suggestions for improving the presentation:**

* It would be helpful to cite the original sources and formally state the results of the baseline methods referenced in Table 1. In its current form, it is not immediately clear that “GD” refers to the results developed in this manuscript.
* In the last sentence of the conclusion,  [1] could be a good reference to include.

[1] Cohen, J., Damian, A., Talwalkar, A., Kolter, Z., & Lee, J.D. (2024). Understanding Optimization in Deep Learning with Central Flows. ArXiv, abs/2410.24206.

**Ethical Concerns:**

["NO or VERY MINOR ethics concerns only"]

**Final Justification:**

Technically interesting work although the presentation could be improved. I keep my score unchanged.

**Limitations:**

See my points in Weaknesses and Questions part.

**Quality:**

3

**Strengths And Weaknesses:**

**Strengths:**
* The manuscript extends an existing line of work to a relevant and practically motivated setting. Unlike the unregularized case, where the minimizer lies at infinity, this setup requires genuinely new technical arguments to handle the $\ell_2$-regularized objective.

**Weaknesses:**
* I find the way the contributions are introduced in the abstract somewhat misleading. While it is true that the proposed method can match Nesterov’s accelerated gradient convergence rate, as discussed in Section 6, this happens only in the regime $\lambda \leq O(n^{-2})$, which is arguably not the most practically relevant case. In contrast, in the more interesting regime $\lambda = O(n^{-1})$, the result yields an iteration complexity of $O(\lambda^{-2/3})$, which remains slower than Nesterov’s method. I would recommend revising the abstract and introduction to clarify this distinction.
* More broadly, although the overall result is slower than a known baseline (Nesterov’s momentum), the paper still offers an interesting contribution given the emerging relevance of edge-of-stability dynamics in practice.
* Finally, the presentation and discussion of the results could be improved for clarity; see my questions and comments below.

---

> ### Author Rebuttal · Authors · 2025-07-29
>
> Thank you for your instructive comments. We will add references for the prior results discussed in Table 1. We address your other questions as follows.
>
> ---
> **Q1.** “I find the way the contributions are introduced in the abstract somewhat misleading. While it is true that the proposed method can match Nesterov’s accelerated gradient convergence rate, … this happens only in the regime $\lambda\le O(n^{-2})$ … I would recommend revising the abstract and introduction to clarify this distinction.”
>
> **A1.** We respectfully disagree that our abstract is misleading. According to the standard definition of big-O notation, $O(\sqrt{\kappa})$ refers to a quantity that grows no faster than $\sqrt{\kappa}$ as $\kappa$ goes to infinity, which corresponds to $\lambda$ goes to zero, everything else being fixed. As the big-O notation is about asymptotics, we do not think there is a misleading statement in our abstract.
>
> In the introduction, we give details on the regime where large stepsize GD matches Nesterov’s acceleration (see the “Contributions” paragraph). We will make this part even clearer by including the precise conditions on $\lambda$. However, we do not think that our abstract or introduction requires significant revision.
>
> ---
> **Q2.** “Both learning rates in Theorem 1 and Theorem 4 require a priori knowledge of the margin parameter gamma. I would expect that if this parameter is not known, the generalization bounds in Table 1 could become suboptimal. Could you discuss how realistic this assumption is in practice? Additionally, it would be helpful to compare the results to adaptive gradient methods (e.g., Zhang et al. [2025]) in the scenario where gamma is unavailable.”
>
> **A2.** You’re correct that the optimal stepsize in Theorems 1 and 4 depends on $\gamma$. However, adaptive GD considered by Zhang et al. [2025] also requires a prior knowledge of $\gamma$ to set the optimal stopping time $1/\gamma\^2$. Therefore our comparison is fair.
>
> In practice, full knowledge of $\gamma$ is not necessary. A lower bound on $\gamma$ (denoted by $\bar\gamma$) together with a validation set are sufficient to determine a near-optimal stepsize. This is because we can set up a multiplicative grid of candidate $\gamma$’s ranging from $\bar\gamma$ to $1$, run GD with the corresponding stepsizes, and choose the optimal run according to its performance on a validation set. Since there are $\Theta(\ln(1/\bar \gamma))$ many candidate $\gamma$’s, this procedure adds at most a logarithmic factor in the total step complexity and in the risk bound.
>
> ---
> **Q3.** “The convergence rate predicted by Theorem 1 is worse than that of Theorem  4 in the range $O(n^{-3/2}) \le \lambda \le O(n^{-1})$, suggesting there may be room for improving the analysis of Theorem 1. What causes this discrepancy?...”
>
> **A3.** This discrepancy exists because we use different approaches to control the stable phase of GD for small and large $\lambda$’s. We refer the reviewer to Lines 249-259 in Section 2.3 for a detailed discussion. We agree that there is room to improve our current analysis, which is a challenging technical problem and would require significant technical innovations upon our current approaches. We will comment on this as an important open problem in the revision.
>
> ---
> **Q4.** “In the last sentence of the conclusion, [1] could be a good reference to include...”
>
> **A4.** We are happy to cite and discuss [1] in our “Related work” section. However, we don’t see a clear connection between the last sentence of our conclusion, which suggests developing mathematical tools to prove the convergence of GD with large stepsizes, and [1], which proposed central flow as a qualitative approximation of large stepsize GD. Due to the existence of a non-vanishing approximation error, central flow does not seem to be directly helpful for our question of rigorously proving the convergence of large stepsize GD. We appreciate further elaboration.

---

> > ### Comment · Reviewer_9WKU · 2025-08-04
> >
> > Thank you for addressing my comments,  I will keep my score.

---

### Decision · Program_Chairs · 2025-09-17

**Decision:**

Accept (poster)

**Comment:**

The paper studies gradient descent with large constant stepsizes for regularized logistic regression with linearly separable data. It shows that acceleration is possible in the small regularization regime, and provides detailed theoretical analysis of the edge-of-stability dynamics.

The proofs develop new arguments not present in earlier unregularized analyses. The work deepens our understanding of large-stepsize dynamics and is of relevance to NeurIPS audience.

The main concerns raised are about practical relevance and presentation. The strongest acceleration result holds under small regularization, which may be less realistic in noisy settings. Several reviewers asked for simple experiments or simulations to illustrate the dynamics. These smaller issues can be addressed when preparing camera ready version.